# EPISTEMIC WRAPPING FOR UNCERTAINTY QUANTIFICATION

## ABSTRACT

Uncertainty estimation is pivotal in machine learning, especially for classification tasks, as it improves the robustness and reliability of models. We introduce a novel 'Epistemic Wrapping' methodology aimed at improving uncertainty estimation in classification. Our approach uses Bayesian Neural Networks (BNNs) as a baseline and transforms their outputs into belief function posteriors, effectively capturing epistemic uncertainty and offering an efficient and general methodology for uncertainty quantification. Comprehensive experiments employing various BNN baselines and an Interval Neural Network for inference on the MNIST, Fashion-MNIST, CIFAR-10 and CIFAR-100 datasets demonstrate that our Epistemic Wrapper significantly enhances generalisation and uncertainty quantification.

## 1 INTRODUCTION

In the realm of machine learning, particularly in classification tasks, uncertainty estimation plays a crucial role in enhancing the robustness and reliability of models (Sale et al., 2023). Accurately quantifying uncertainty is vital for applications where decisions must be made with confidence, such as in medical diagnosis (Lambrou et al., 2010), autonomous driving (Fort & Jastrzebski, 2019) and financial forecasting. Traditional deterministic neural networks, while powerful, cannot often effectively capture and express uncertainty (Liu et al., 2020). This shortfall has spurred interest in probabilistic approaches, with Bayesian neural networks (BNNs) emerging as a promising solution in this context. BNNs offer a principled approach to uncertainty estimation by incorporating prior distributions over the model parameters, leading to posterior distributions that reflect model uncertainty (Jospin et al., 2022). Despite their theoretical appeal, BNNs face practical challenges, including high computational costs and complexity in training.

The literature majors on two sources of uncertainty: *Epistemic* Uncertainty (EU) and *Aleatoric* Uncertainty (AU) (Hüllermeier & Waegeman, 2021; Abdar et al., 2021). Epistemic uncertainty is due to a lack of knowledge about the true model parameters and can be reduced with more data or better models. In contrast, aleatoric uncertainty stems from the inherent randomness in the data generation process and cannot be reduced. Over the years, various studies (Hüllermeier & Waegeman, 2021; Abdar et al., 2021) have recognised that accurately modelling parameter uncertainty can produce a variety of credible network models, which are likely to include the true underlying network model, leading to both better EU estimation and more reliable inference. In particular, *second-order uncertainty* frameworks (including belief functions Cuzzolin (2020)) can be employed to model both EU and AU, effectively expressing 'uncertainty about a prediction's uncertainty' (Hüllermeier & Waegeman, 2021; Sale et al., 2023).

BNNs, as one of the prevalent method for uncertainty estimation, treat all the weights and biases of the network as probability distributions. The prediction of the Neural Network is represented as a second-order distribution, thus representing the probability distribution of distributions (Hüllermeier & Waegeman, 2021). Although effective approximation techniques have been developed, such as variational inference (VI) approaches (Blundell et al., 2015; Gal & Ghahramani, 2016) and sampling methods (Neal et al., 2011; Hoffman et al., 2014), the high computational cost of BNNs during training as well as inference time limit their practical adoption, especially in real-time applications (Abdar et al., 2021).

Recent work shows that epistemic uncertainty (EU) can be better captured by frameworks more general than probability distributions Cuzzolin (2024), including credal sets Levi (1980) and belief

functions Shafer (1976), leading to improved robustness and uncertainty estimation Manchingal & Cuzzolin (2022); Manchingal et al. (2025b; 2023); Chan et al. (2024); Wang et al. (2024b;a); Caprio et al. (2024); Manchingal et al. (2025a).

Still, current efforts model (epistemic) uncertainty in the model's *target* space, rather than its *parameter* space.

This paper proposes *Epistemic Wrapper*, a novel method which, for the first time, models EU in the parameter space via a random set representation by 'wrapping' a learnt Bayesian posterior in the form of a *belief function* (Fig. 1). The methodology unfolds organically, beginning with a Bayesian Neural Network (BNN) and culminating in an Interval Neural Network (INN). From a pre-trained BNN, we obtain posterior distributions with parameters $(\mu, \sigma)$, where Gaussian priors are assumed. These posterior distributions are truncated and we compute continuous belief functions over closed intervals. Through Möbius inversion, these belief functions translate into corresponding mass values. These discrete, normalized mass values are then embedded into a continuous representation by fitting a Dirichlet distribution with parameters estimated via the method of L-moments. The interval-valued nature of Dirichlet sampling motivates the use of an Interval Neural Network (INN), which performs stable and calibrated inference. To our knowledge, no previous work has modelled EU in the parameter space via higher-order uncertainty measures. The motivation behind modelling epistemic uncertainty in the parameter space stems from the idea that parameter uncertainty is a primary source of EU. By representing uncertainty directly in the parameter space before it propagates to predictions,

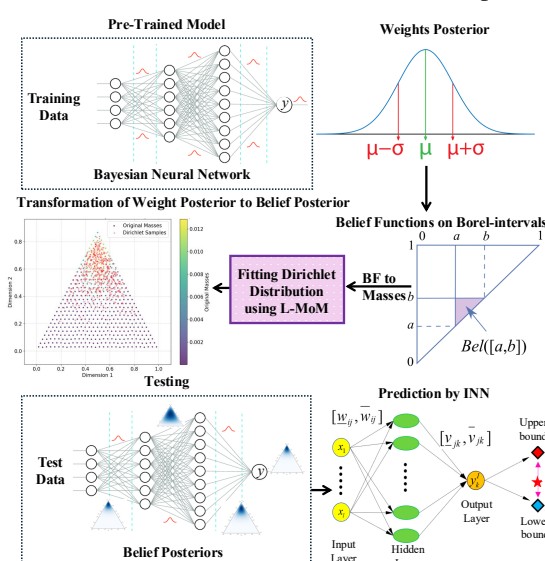

Figure 1: **Epistemic Wrapper** transforms weights posteriors from a Bayesian Neural Network into belief posteriors. It involves extracting probability posteriors, calculating belief values over Borel intervals, computing mass values using Moebius inversion, fitting a Dirichlet distribution to these masses via method of L-moments, and using the resulting belief posteriors as weights to Interval Neural Networks for final predictions.

we capture model-level uncertainty in a more principled way. Modelling in the parameter space offers several advantages: (a) It provides a prior-agnostic mechanism to represent epistemic uncertainty, without relying solely on the model's output distribution. (b) It enables structured and interpretable sampling through belief functions and Dirichlet distributions, supporting more stable and calibrated uncertainty estimates via interval-based inference. (c) It can be seamlessly integrated with existing BNNs offering flexibility and broader applicability.

Although Bayesian neural networks do, in theory, model epistemic uncertainty in the parameter space through the posterior, practical implementations rely on a *single* approximating distribution (e.g., a factorised Gaussian in variational inference or the implicit posterior of an ensemble). These are first-order representations and do not provide any second-order structure such as credal sets or random sets. Our approach differs precisely in this respect: the Epistemic Wrapper does not replace the BNN posterior, but *enriches* it by constructing a belief-function representation that yields a set of plausible posteriors rather than a single one. This produces a genuine second-order (random-set) model of epistemic variability in parameter space. Existing methods, including evidential deep learning and ensembles, operate on the predictive distribution or rely on a single posterior approximation, but do not introduce this higher-order parameter-space structure. The distinction is therefore not whether epistemic uncertainty originates in the parameter space it does but *how* it is represented. Our method constructs interval-based belief and plausibility bounds over parameters before prediction, offering a more expressive and robust characterisation of epistemic uncertainty. To the best of our knowledge, this is the first approach to build such a higher-order epistemic representation directly over BNN parameters, enabling principled model-level uncertainty quantification.

The strength of our approach lies in the novel, careful and coherent integration of theoretically-grounded techniques. While individual components, such as belief function construction and Dirichlet fitting, are based on existing concepts, their combination into a framework that learns Random Set (RS) representations directly over BNN parameters is new and forms the core contribution of this work. In particular, the likelihood transformation at the core of our approach is not heuristic but is grounded in a set of rigorous rationality axioms. It is in fact the only belief function satisfying these properties, which provides a formal and well-founded basis for this transformation. The subsequent use of the Dirichlet mass function to encode the continuous belief function is equally principled. Our choice is guided by the need for an efficient RS representation on the collection of intervals, where any probability distribution defined over the simplex could theoretically be used. However, Dirichlet distributions have recently demonstrated strong practical effectiveness in modelling epistemic uncertainty in neural networks, as highlighted in evidential learning literature Sensoy et al. (2018). Finally, inference with INNs is also statistically motivated, as it provides an efficient way to propagate belief-based uncertainty into interval predictions. This ensures that every element of our wrapping framework is anchored in rigorous theoretical principles, rather than assembled heuristics.

Our approach leverages the strengths of BNNs while injecting the ability of higher-order measure to improve robustness and uncertainty estimation. The contributions to the literature are therefore: (1) **The first modelling of EU in the parameter space using higher-order uncertainty measures**. (2) A novel and versatile **Epistemic Wrapper** concept, that can be applied to any BNN baseline to convert it automatically into a belief-function posterior. (3) Based on the above, a **novel approach to uncertainty estimation** in classification which efficiently leverages BNNs as a foundation. Our experiments further demonstrate the versatility of the proposed Epistemic Wrapper across multiple datasets. For example, on MNIST, the baseline BNN achieved an accuracy of $72.44\% \pm 0.24$, whereas Epi-Wrapper substantially improved performance to $91.02\% \pm 0.05$. On Fashion-MNIST, the BNN reached $58.91\% \pm 0.24$, while Epi-Wrapper achieved $82.45\% \pm 0.10$. Similar improvements are consistently observed on large-scale benchmarks such as CIFAR-10 and CIFAR-100 with ResNet-18 and VGG-16 backbones, highlighting the effectiveness and scalability of our approach in enhancing predictive performance and reliability across diverse settings.

## 2 RELEVANT WORK

**Epistemic approaches.** While various types of uncertainty measures Cuzzolin (2021) have been employed in machine learning in the past Cuzzolin & Gong (2013); Cuzzolin (2018a); Liu et al. (2019); Gong & Cuzzolin (2017), recent advancements in epistemic uncertainty modelling have introduced a range of methods to improve predictive reliability across various neural architectures. Evidential deep learning predicts second-order probability distributions to estimate uncertainty, but faces challenges in optimisation and interpretation (Juergens et al., 2024). Methods like G-$\Delta$UQ refine uncertainty calibration in Graph Neural Networks (GNNs) through stochastic data centring (Trivedi et al., 2024), while Stochastic Partial Differential Equation (SPDE)-based GNNs employ $Q$-Wiener processes for uncertainty propagation in complex graphs (Lin et al., 2024). The Graph Energy-Based Model (GEBM) leverages graph diffusion to quantify uncertainty at different structural levels (Fuchsgruber et al., 2024), and credal set-based ensemble learning constructs plausible probability distributions to measure aleatoric and epistemic uncertainty (Hofman et al., 2024).

Crucially, (Manchingal et al., 2025a) introduces a unified evaluation framework for uncertainty-aware classifiers, mapping all uncertainty-aware predictions into credal sets Cuzzolin (2008a), thus enabling a standardised assessment of epistemic uncertainty across BNNs, Deep Ensembles, Evidential Deep Learning (EDL), and Credal Set-based approaches. (Manchingal et al., 2025b) extends uncertainty modelling through Random-Set Neural Networks (RS-NNs), which employ random set theory to construct belief-based uncertainty representations, providing a more flexible alternative to conventional probabilistic models. Credal Interval Neural Networks Wang et al. (2025), instead, represent predictions as credal sets, which encapsulate a range of probable outcomes, thereby explicitly modelling epistemic uncertainty. Building on the latter, Credal Deep Ensembles (Wang et al., 2024b) predict and aggregate ensembles of convex sets of probability distributions, resulting in a more conservative and informative epistemic uncertainty quantification. In an alternative approach Charpentier et al. (2020); Malinin et al. (2019); Sensoy et al. (2018) predictions are modelled as

Dirichlet distributions. A key challenge with these methods is the lack of ground truth labels for uncertainty, making direct supervision difficult.

While these models can be highly effective, they primarily quantify uncertainty at the target level, leaving the question of modelling epistemic uncertainty at parameter level open. In contrast, our proposed Epistemic Wrapper leverages BNNs to do exactly so, by transforming probability posteriors into belief posteriors, to offer a robust solution for uncertainty quantification in classification tasks.

**Interval neural networks.** Traditional *Interval Neural Networks* (INNs) employ deterministic interval-based representations for inputs, outputs, weights, and biases, ensuring robust uncertainty modelling in neural computations. The forward propagation in an INN follows interval arithmetic principles, where the interval-formed activations in each layer are computed using element-wise interval addition, subtraction, and multiplication (Hickey et al., 2001). Specifically, the activation output of the $l^{\text{th}}$ layer is determined by applying a monotonically increasing activation function to the interval-weighted sum of the previous layer's outputs and the corresponding interval biases. This formulation guarantees the *set constraint* property, ensuring that for any given input and network parameters within their defined intervals, the computed activations remain bounded within a well-defined range. When the activation function is non-negative (e.g., ReLU), further simplifications allow efficient computation of interval bounds using minimum and maximum operators. This structured interval propagation enables INNs to maintain rigorous mathematical constraints while modelling uncertainties in deep learning architectures (Morales & Sheppard, 2025).

Our approach employs INNs at inference time using our epistemic wrapper weights, sampled from the wrapped belief posterior, thus leveraging their structured interval-based representations to quantify and propagate epistemic uncertainty effectively.

## 3 METHODOLOGY

**Overview**: Before presenting the full technical construction, we briefly outline the intuition behind the Epistemic Wrapper. A Bayesian neural network provides a point-valued posterior density over each weight. The wrapper enriches this by constructing a bounded random-set representation, summarising the posterior through belief and plausibility bounds. These bounds are then mapped to a Dirichlet distribution and propagated through an Interval Neural Network for inference. The following steps make this pipeline explicit. A complete step-by-step pseudo-code implementation of the Epistemic Wrapper is provided in Appendix B.1.

### 3.1 LEARNING A BAYESIAN POSTERIOR (BASELINE)

*Intuition.* The wrapper begins with a standard variational BNN, since it provides the posterior density we later convert into a random-set representation.

Epistemic wrapping begins by exploiting BNNs with Gaussian priors parameterized by $(\mu_0, \sigma_0)$; training then yields posterior distributions over the weights, also parameterized by $(\mu, \sigma)$. The posterior distribution $p(\boldsymbol{\omega}|\mathbb{D})$ is defined via Bayes' theorem, but is generally intractable. To address this, we employ Variational Inference, approximating $p(\boldsymbol{\omega}|\mathbb{D})$ with a variational distribution $q(\boldsymbol{\omega})$ that is optimized to match the true posterior. At inference time, Bayesian Model Averaging is performed by sampling weights from the variational posterior.

### 3.2 DYNAMIC TRUNCATION

*Intuition.* Since belief functions require bounded sets, we first map each posterior density to a compact interval that adapts to the scale of the learned variance. A typical Bayesian weight posterior will look like the one in Fig. 2. These posterior distributions are then truncated using a *dynamic distribution truncation* mechanism, an adaptive technique that defines the range of a distribution based on its mean and standard deviation. This dynamically scales the bounds to the parameter values according to the variance of the distribution, ensuring tighter truncation for distributions with smaller variances, where the probability mass is more concentrated, and looser bounds for those with larger variances. The truncation bounds are calculated as: Lower Bound $= \mu - \text{dynamic\_multiplier} \cdot$

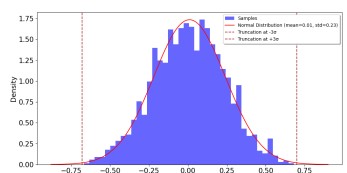

Figure 2: Posterior distribution of last-layer weights with truncation at $\pm 3\sigma$.

$\sigma$, Upper Bound $= \mu + \text{dynamic\_multiplier} \cdot \sigma$, where $\mu$ is the mean, $\sigma$ is the standard deviation, and dynamic\_multiplier $= \min(5.0, \frac{1.0}{\sigma})$, ensuring that the multiplier decreases for low-variance distributions while capping its value at 5.0 to prevent excessive truncation in high-variance cases. We selected this approach as it provides a balance between capturing the significant probability mass of the distribution and avoiding overly wide or narrow bounds, which could either dilute meaningful mass representation or exclude critical probabilistic regions.

### 3.3 CONTINUOUS BELIEF FUNCTIONS ON CLOSED INTERVALS

*Intuition.* The truncated posterior is then lifted to a belief function, providing lower and upper probability bounds that quantify epistemic imprecision in the parameter space.

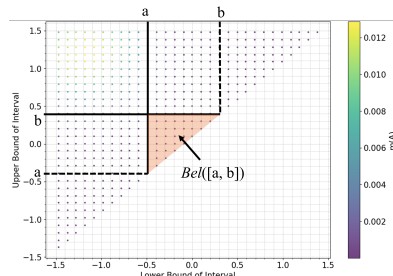

Belief functions (Cuzzolin, 2014b) can be easily extended to continuous spaces (e.g., a network's parameter space) by defining a continuous mass function over the collection of *closed intervals*, rather than the entire power set. For an introduction to belief functions, see Appendix Section A.1. Given a network parameter $\omega$ with values in $\mathbb{R}$, this requires defining a continuous PDF over the collection of intervals $[a, b] \subset \mathbb{R}$ (Cuzzolin, 2020). Here we will assume that parameter values are bounded after truncation (for illustration, in $[0, 1]$); however, the method can be easily extended to unbounded parameter values as well. The space of all closed intervals in $[0, 1]$ is a triangle, as illustrated in Fig. 3. Given a continuous mass function there (non-negative and with integral 1), one

Figure 3: Graphical visualisation of the continuous PDF/mass function over intervals, with the area representing Bel($[a, b]$).

can compute the belief and plausibility value of a parameter interval $A = [a, b]$ by integrating it over specific regions of the triangle Smets (2005) (Fig. 3). The same applies for parameters bounded by arbitrary values.

Given a truncated posterior distribution over a network's weights, learned by a BNN, our Wrapper transforms it into a continuous belief function using the method proposed in Wasserman (1990). For any closed interval $A = [a, b]$ of the parameter space, one can compute its *plausibility* from the posterior distribution by taking the supremum of the normalised posterior $\hat{p}(\omega|\mathbb{D})$ across all $\omega \in A$, namely:

$$Pl_\Theta(A|\mathbb{D}) = \sup_{\omega \in A} \hat{p}(\omega|\mathbb{D}). \tag{1}$$

The corresponding belief value is then calculated as the complement of the plausibility:

$$Bel_\Theta(A|\mathbb{D}) = 1 - Pl_\Theta(A^c|\mathbb{D}), \tag{2}$$

ultimately providing the sought random-set representation in the parameter space.

The method is grounded into rationality principles, such as (i) the likelihood principle, (ii) compatibility with Bayesian inference (which ensures that combining a Bayesian prior with the belief function yields the Bayesian posterior), and (iii) the principle of Minimum Commitment, which maintains that among the belief functions satisfying the previous two principles, the one chosen should commit to the least amount of information necessary Cuzzolin (2020).

To cap complexity, sample belief values can be computed for a grid of parameter values only. The corresponding mass values can then be easily obtained by Moebius inversion (Shafer, 1976).

### 3.4 FITTING A DIRICHLET DISTRIBUTION

*Intuition.* Because the Moebius mass values lie on a probability simplex, we model them with a Dirichlet distribution, using L-moments for stable parameter estimation.

Epistemic Wrapper employs the method of *L-moments* Hosking (2018) to fit a Dirichlet distribution to the grid of mass values so obtained.

A **Dirichlet distribution** is a family of continuous multivariate probability distributions parameterised by a vector $\alpha$ of positive real numbers; in fact, a multivariate extension of the Beta distribution

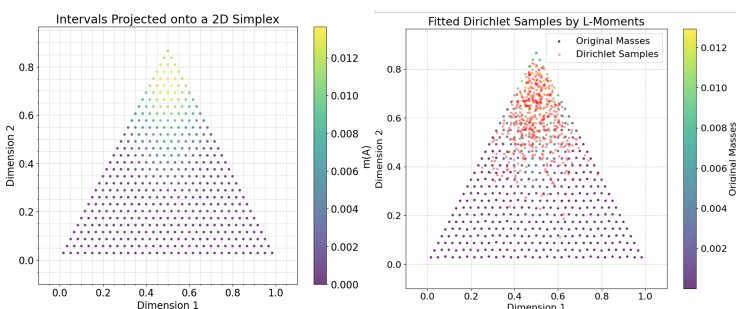

Figure 4: **Left:** Intervals projected onto a 2D simplex. Each point represents an interval $A = [a, b]$ with its location determined by the values $a$ and $b$, and the colour scale indicates the corresponding mass values $m(A)$, ranging from 0.00 to 0.012. **Right:** Visualization of Dirichlet samples on a 2D simplex. Points sampled from the fitted Dirichlet distribution over mass values.

$$f(x_1, \ldots, x_K; \alpha_1, \ldots, \alpha_K) = \frac{1}{B(\alpha)} \prod_{i=1}^{K} x_i^{\alpha_i - 1}. \tag{3}$$

As they are defined on the collection of vectors $x \in [0, 1]^K$ of dimension $K$ whose coordinates add to 1, Dirichlet distributions can be interpreted as second-order distributions. For an introduction to Dirichlet distribution, see Appendix Section A.2.

The **method of L-Moments** is a statistical approach employed for parameter estimation in probability distributions. Here we utilise this method to estimate the parameters of a Dirichlet distribution over mass values. L-moments are analogous to conventional moments but are based on linear combinations of order statistics.

To fit a Dirichlet distribution to the grid of mass values, we compute weighted L-moments from the data represented in a 3D simplex space, where each data point has an associated weight derived from its mass value. An example grid in a 2D simplex representation is shown in Fig. 4.

**Computation of weighted L-Moments**. We first need to compute the first-order and second-order weighted L-moments from the grid of data points. Let $\mathbf{x}_i \in \mathbb{R}^3$ denote the $i$-th data point in the 3D simplex and $w_i$ its associated weight (derived by normalizing the mass values, so that $\sum_i w_i = 1$). The L-moments are computed as follows. *First-order L-moment ($L_1$).* The weighted mean of the points in the simplex and is given by:

$$L_1 = \sum_{i=1}^{n} w_i \mathbf{x}_i. \tag{4}$$

*Second-order L-moment ($L_2$).* The weighted spread (variance) of the points relative to $L_1$:

$$L_2 = \frac{\sum_{i=1}^{n} w_i (\mathbf{x}_i - L_1)^2}{\sum_{i=1}^{n} w_i} \tag{5}$$

To ensure numerical stability, a small value $\epsilon$ is added to $L_2$ when necessary, preventing division by zero in subsequent computations.

Using the computed L-moments, we can estimate the parameters $\boldsymbol{\alpha} = (\alpha_1, \alpha_2, \alpha_3)$ of the Dirichlet distribution. The relationship between L-moments and the Dirichlet parameters is expressed as:

$$\alpha_k = L_{1,k} \left( \frac{L_{1,k}(1 - L_{1,k})}{L_{2,k}} - 1 \right), \quad k = 1, 2, 3 \tag{6}$$

where $L_{1,k}$ and $L_{2,k}$ are the respective components of the first and second L-moments along each axis of the simplex. Note that $k = 3$ corresponds to the dimensionality of the projected probability simplex. Although equation 6 can, in principle, yield negative $\alpha_k$ values if the second L-moment $L_{2,k}$ exceeds $L_{1,k}(1 - L_{1,k})$ (which may occur in sparse or highly skewed data), our implementation prevents this in practice. We clamp unstable $L_2$ values, enforce a small positive threshold $\epsilon$ on all

$\alpha_k$. These safeguards ensure that the estimated Dirichlet parameters remain valid, positive, and numerically stable across datasets.

Visual representation show that, after fitting a Dirichlet distribution to the grid of mass values, samples of it are also concentrated on the top of the simplex as shown in Fig. 4-Right.

**Theoretical Properties of the Epistemic Wrapper**. The Epistemic Wrapper preserves an important theoretical property. Specifically, the original Bayesian posterior $P$ lies within the credal set induced by the belief and plausibility functions after wrapping, satisfying

$$Bel(A) \leq P(A) \leq Pl(A) \quad \text{for all measurable sets } A.$$

This relation ensures that our transformation is conservative: it enriches the original posterior with second-order uncertainty without distorting the underlying predictive information. Consequently, the model maintains consistency with the Bayesian posterior while gaining robustness, which helps to explain the observed improvements in generalization and uncertainty estimation. The plausibility $Pl(A)$ captures the maximum value of $\hat{p}(\omega|\mathbb{D})$ over $A$, while belief $Bel(A)$ captures the minimum guaranteed mass by considering the complement $A^c$. Since $P(A)$ is the integral of $\hat{p}(\omega|\mathbb{D})$ over $A$, it must lie between the least conservative estimate (Bel) and the most generous estimate (Pl) over $A$. This follows from the construction rules of likelihood-based belief functions and random set theory (see (Shafer, 1976; Wasserman, 1990; Cuzzolin, 2020)).

### 3.5 INFERENCE VIA INTERVAL NEURAL NETWORKS

*Intuition.* Dirichlet samples define weight intervals, which align naturally with the interval arithmetic of an INN and allow us to propagate epistemic uncertainty through the network.

Since Dirichlet sampling yields interval-based representations, the framework integrates naturally with an Interval Neural Network (for details see Appendix A.3), where weight intervals are derived from a combination of Dirichlet-based intervals (wrapped parameters) and Gaussian posteriors (unwrapped parameters). While the overall architecture follows a standard INN, our formulation integrates both Dirichlet- and Gaussian-based uncertainty representations within a unified framework. This hybrid design enables stable and well-calibrated uncertainty estimates, particularly for epistemic uncertainty, and ensures smooth interaction between wrapped and unwrapped weights during inference. The unwrapped weights produced by Gaussian posteriors generate the standard bounds

$$\underline{\omega} = \mu - \sigma, \qquad \overline{\omega} = \mu + \sigma.$$

In contrast, wrapped weights obtained from the Epistemic Wrapper use interval bounds induced by Dirichlet samples (see Appendix A.4). Together, these two forms of interval weights constitute the INN's initialisation at inference time (Kim, 1993). The baseline INN uses only Gaussian-derived intervals, while the Epi-Wrapper replaces them with wrapped and unwrapped weights. Both models are subsequently fine-tuned under identical optimisation settings.

## 4 EXPERIMENTS

In this section, we evaluate the effectiveness of our Epistemic Wrapper in terms of uncertainty estimation and predictive performance. We summarize the experimental setup, including datasets, model architectures, and ablation studies, and compare our method against relevant baselines. We employed Bayesian baselines including BNNR (Auto-Encoding Variational Bayes (Kingma & Welling, 2013) with the local re-parameterization trick (Molchanov et al., 2017)), and BNNF (Flipout gradient estimator with the negative evidence lower bound loss (Wen et al., 2018)). We use four standard image classification benchmarks: MNIST (LeCun, 1998), Fashion-MNIST (Xiao et al., 2017), CIFAR-10, and CIFAR-100 (Krizhevsky, 2009b). Further implementation details of the Epi-Wrapper algorithm are provided in Appendix B.

### 4.1 BUDGETING

A budgeting strategy is introduced to selectively transform the posterior distributions of a *subset* of parameters (weights and biases). Posteriors not selected, referred to as *unwrapped* posteriors, retain their original learned parameters. We propose four distinct budgeting strategies: three are parameter-based, prioritizing posteriors with high $\mu$, high $\sigma$, or simultaneously high $\mu$ and $\sigma$, while the fourth employs a random selection strategy that remains unbiased w.r.t. these parameter values.

Table 1: Classification accuracies on MNIST under different budgeting criteria before and after fine-tuning for posterior weights. Results are from 15 runs. Best scores are presented in bold.

| BUDGETING | MLP SIZE | BEFORE FINE-TUNING | | AFTER FINE-TUNING | | |
|---|---|---|---|---|---|---|
| | | INN | EPI-WRAPPER | BNN | INN | EPI-WRAPPER |
| $\uparrow \sigma$ | 2 | $9.11 \pm 0.53$ | $\mathbf{12.93 \pm 0.75}$ | $33.14 \pm 0.22$ | $57.77 \pm 0.81$ | $\mathbf{62.43 \pm 0.46}$ |
| | 4 | $10.44 \pm 0.77$ | $\mathbf{19.94 \pm 0.47}$ | $40.19 \pm 0.55$ | $83.32 \pm 0.24$ | $\mathbf{85.17 \pm 0.10}$ |
| | 8 | $9.33 \pm 0.54$ | $\mathbf{25.46 \pm 1.57}$ | $72.44 \pm 0.24$ | $\mathbf{91.12 \pm 0.08}$ | $91.08 \pm 0.09$ |
| $\uparrow \mu$ | 2 | $9.11 \pm 0.53$ | $\mathbf{10.63 \pm 0.34}$ | $33.14 \pm 0.22$ | $57.77 \pm 0.81$ | $\mathbf{63.06 \pm 0.47}$ |
| | 4 | $10.44 \pm 0.77$ | $\mathbf{18.13 \pm 0.71}$ | $40.19 \pm 0.55$ | $83.32 \pm 0.24$ | $\mathbf{85.35 \pm 0.06}$ |
| | 8 | $9.33 \pm 0.54$ | $\mathbf{51.33 \pm 1.21}$ | $72.44 \pm 0.24$ | $\mathbf{91.12 \pm 0.08}$ | $91.02 \pm 0.05$ |
| $\uparrow \mu + \sigma$ | 2 | $9.11 \pm 0.53$ | $\mathbf{10.45 \pm 0.16}$ | $33.14 \pm 0.22$ | $57.77 \pm 0.81$ | $\mathbf{63.02 \pm 0.55}$ |
| | 4 | $10.44 \pm 0.77$ | $\mathbf{18.55 \pm 0.68}$ | $40.19 \pm 0.55$ | $83.32 \pm 0.24$ | $\mathbf{85.18 \pm 0.07}$ |
| | 8 | $9.33 \pm 0.54$ | $\mathbf{51.31 \pm 1.29}$ | $72.44 \pm 0.24$ | $91.12 \pm 0.08$ | $\mathbf{91.12 \pm 0.07}$ |
| RANDOM-SELECTION | 2 | $\mathbf{9.11 \pm 0.53}$ | $9.80 \pm 0.00$ | $33.14 \pm 0.22$ | $57.77 \pm 0.81$ | $\mathbf{64.84 \pm 0.16}$ |
| | 4 | $10.44 \pm 0.77$ | $\mathbf{17.35 \pm 0.27}$ | $40.19 \pm 0.55$ | $83.32 \pm 0.24$ | $\mathbf{85.45 \pm 0.06}$ |
| | 8 | $\mathbf{9.33 \pm 0.54}$ | $9.23 \pm 0.64$ | $72.44 \pm 0.24$ | $\mathbf{91.12 \pm 0.08}$ | $90.80 \pm 0.09$ |

**Ablation on Budgeting**. We first conducted an ablation study on the MNIST dataset in which four different Budgeting criterias were tested.

In **Budgeting using High Variance** ($\uparrow \sigma$) we sampled $5\%$ weights with 'High Variance' from the posterior distributions (parameters: $\mu, \sigma$) of the whole model and transformed them to belief posteriors using Epistemic Wrapper. The results are shown in Table 1, where 'MLP size' is the number of hidden units in the single hidden layer of the model. Since inference in our methodology is done using INNs, we compare our results with those of INN (taken as a baseline). The results shows that using the wrapper improves the quality of the weights initialization with respect to the INN baseline. For instance, an MLP with 32 hidden units and weights randomly initialized achieved an accuracy of $10.37\%$ on the test data, while for our wrapper the test accuracy was $50.20\%$.

**Budgeting using High Mean** ($\uparrow \mu$) is another strategy in which we sample and 'wrap' the $5\%$ weights with 'High Mean' from the posterior distributions. From the results shown in Table 1, it can be seen that 'High Mean' performs better for MLP size (no hidden units) $= 8$.

**In Budgeting using High Mean and High Variance** ($\uparrow (\mu, \sigma)$) we rank the parameters by computing a combined score, defined as the sum of the mean and variance of their posterior distributions: combined_score $= \mu + \sigma$. This acts as a proxy for an upper bound of the posterior distribution, allowing us to prioritize parameters that are either highly informative (high mean) or uncertain (high variance). We then wrap these top $5\%$ weights using Epi-wrapper. The results are shown in Table 1. This strategy allows us to selectively wrap the most influential and uncertain parameters, ensuring that the transformation captures meaningful epistemic uncertainty. However, this approach also imposes a strict constraint on the selection process, as only weights satisfying both conditions are chosen, which may limit flexibility in certain scenarios.

**Budgeting using Random Selection** ($\mu, \sigma$) is done by randomly selecting $5\%$ weights from the baseline BNN and extract belief posteriors using the wrapper. Table 1 shows that the results are worse than with other strategies. This is due to the fact that random sampling, while giving us an unbiased selection of posterior weights, may miss those posterior distributions with high uncertainty that can be improved using our wrapping approach.

**Note:** The lower MNIST and Fashion-MNIST accuracies come from the very small Bayesian backbone we used: a single-layer variational MLP with only 8 hidden units.

### 4.2 FINE-TUNING

We performed the fine-tuning of the models, the INN (baseline) and Epi-Wrapper (ours), on the training data. The results are shown in Table 1 and Table 2. In comparison to the INN and BNN baselines, our model performs well as the wrapping of weights acts as an initialization strategy in fine-tuning. The details of Fine-Tuning are presented in Appendix Section C.2

### 4.3 ID AND OOD EXPERIMENTAL EVALUATION

To evaluate how well the model captures epistemic uncertainty, we consider both *in-distribution* (iD) and *out-of-distribution* (OoD) datasets. The iD dataset corresponds to the standard test split of

Table 2: Classification accuracies of INN and Epi-Wrapper models before and after fine-tuning across multiple datasets and Bayesian backbones (MLP, LeNet-5, ResNet-18, VGG-16) with BNNF and BNNR baselines. Reported values are mean $\pm$ standard deviation over 15 runs.

| DATASET | BACKBONE | # PARAMS | BASELINE | BNN | BEFORE FINE-TUNING | | AFTER FINE-TUNING | |
|---|---|---|---|---|---|---|---|---|
| | | | | | INN | EPI-WRAPPER | INN | EPI-WRAPPER |
| MNIST | MLP | 12.7K | BNNF | $72.44 \pm 0.24$ | $9.33 \pm 0.54$ | **$51.33 \pm 1.21$** | $91.07 \pm 0.08$ | **$91.12 \pm 0.05$** |
| FASHION-MNIST | MLP | 12.7K | BNNF | $58.91 \pm 0.24$ | $8.57 \pm 1.03$ | **$26.93 \pm 1.44$** | $82.41 \pm 0.19$ | **$82.45 \pm 0.10$** |
| CIFAR-10 | LENET-5 | 166.3K | BNNF | $47.26 \pm 0.24$ | $9.80 \pm 0.18$ | **$42.34 \pm 0.12$** | $45.92 \pm 0.36$ | **$47.89 \pm 0.01$** |
| | | | BNNR | $47.09 \pm 0.13$ | $9.80 \pm 0.18$ | **$42.45 \pm 0.20$** | $45.92 \pm 0.36$ | **$47.99 \pm 0.09$** |
| | RESNET-18 | 9.82M | BNNF | $86.79 \pm 0.21$ | $10.38 \pm 0.17$ | **$20.11 \pm 0.02$** | $87.07 \pm 0.23$ | **$90.28 \pm 0.09$** |
| | | | BNNR | $85.83 \pm 0.30$ | $10.38 \pm 0.17$ | **$26.71 \pm 0.11$** | $87.07 \pm 0.23$ | **$89.96 \pm 0.07$** |
| | VGG-16 | 30.24M | BNNF | $87.39 \pm 0.67$ | $9.56 \pm 0.33$ | **$65.83 \pm 0.06$** | $87.99 \pm 0.30$ | **$88.70 \pm 0.06$** |
| | | | BNNR | $88.29 \pm 0.87$ | $9.56 \pm 0.33$ | **$50.31 \pm 0.08$** | $87.99 \pm 0.30$ | **$89.87 \pm 0.09$** |
| CIFAR-100 | RESNET-18 | 9.8M | BNNF | $67.38 \pm 0.92$ | $1.23 \pm 0.11$ | **$13.29 \pm 0.51$** | $62.70 \pm 0.19$ | **$67.41 \pm 0.33$** |
| | | | BNNR | $67.47 \pm 0.45$ | $1.23 \pm 0.11$ | **$19.48 \pm 0.05$** | $62.70 \pm 0.19$ | **$67.49 \pm 0.32$** |
| | VGG-16 | 30.24M | BNNF | $65.48 \pm 0.85$ | $0.96 \pm 0.04$ | **$23.32 \pm 0.04$** | $60.10 \pm 1.04$ | **$65.55 \pm 0.13$** |
| | | | BNNR | $66.59 \pm 0.36$ | $0.96 \pm 0.04$ | **$16.11 \pm 0.05$** | $60.10 \pm 1.04$ | **$66.63 \pm 0.19$** |

Table 3: OoD detection results (AUROC, AUPRC, and mean entropy) for INN (baseline) and Epi-Wrapper (ours). Models are trained on in-distribution (iD) datasets (MNIST, CIFAR–10, CIFAR–100) and evaluated on their corresponding out-of-distribution (OoD) datasets: MNIST $\rightarrow$ Fashion-MNIST, CIFAR–10 $\rightarrow$ SVHN, CIFAR–100 $\rightarrow$ TinyImageNet. Results are reported for BNNF and BNNR baselines using MLP, LeNet–5, ResNet–18, and VGG–16 backbones.

| DATASET | BACKBONE | # PARAMS | BASELINE | AUROC ($\uparrow$) | | AUPRC ($\uparrow$) | | ENTROPY ($\uparrow$) | |
|---|---|---|---|---|---|---|---|---|---|
| | | | | INN | EPI-WRAPPER | INN | EPI-WRAPPER | INN | EPI-WRAPPER |
| MNIST | MLP | 12.7K | BNNF | $0.532 \pm 0.011$ | **$0.667 \pm 0.009$** | $0.895 \pm 0.035$ | **$0.912 \pm 0.055$** | $0.201 \pm 0.044$ | **$0.287 \pm 0.067$** |
| FASHION-MNIST | MLP | 12.7K | BNNF | $0.605 \pm 0.041$ | **$0.690 \pm 0.076$** | $0.855 \pm 0.015$ | **$0.922 \pm 0.015$** | $0.247 \pm 0.030$ | **$0.299 \pm 0.019$** |
| CIFAR-10 | LENET-5 | 166.3K | BNNF | $0.678 \pm 0.006$ | **$0.749 \pm 0.001$** | $0.620 \pm 0.048$ | **$0.751 \pm 0.003$** | $1.763 \pm 0.010$ | **$1.995 \pm 0.045$** |
| | | | BNNR | $0.678 \pm 0.006$ | **$0.781 \pm 0.009$** | $0.620 \pm 0.004$ | **$0.728 \pm 0.007$** | $1.763 \pm 0.010$ | **$2.005 \pm 0.013$** |
| | RESNET-18 | 9.82M | BNNF | $0.806 \pm 0.014$ | **$0.859 \pm 0.003$** | $0.724 \pm 0.019$ | **$0.799 \pm 0.008$** | $0.569 \pm 0.035$ | **$0.659 \pm 0.034$** |
| | | | BNNR | $0.806 \pm 0.014$ | **$0.861 \pm 0.005$** | $0.724 \pm 0.019$ | **$0.824 \pm 0.008$** | $0.569 \pm 0.035$ | **$0.764 \pm 0.021$** |
| | VGG-16 | 30.24M | BNNF | $0.849 \pm 0.011$ | **$0.850 \pm 0.011$** | $0.796 \pm 0.014$ | **$0.801 \pm 0.015$** | $0.606 \pm 0.058$ | **$0.611 \pm 0.046$** |
| | | | BNNR | $0.849 \pm 0.011$ | **$0.856 \pm 0.010$** | $0.796 \pm 0.014$ | $0.799 \pm 0.015$ | **$0.606 \pm 0.058$** | $0.601 \pm 0.034$ |
| CIFAR-100 | RESNET-18 | 9.8M | BNNF | $0.616 \pm 0.031$ | **$0.705 \pm 0.089$** | $0.688 \pm 0.005$ | **$0.770 \pm 0.040$** | $1.908 \pm 0.043$ | **$1.912 \pm 0.067$** |
| | | | BNNR | $0.616 \pm 0.031$ | **$0.690 \pm 0.019$** | $0.688 \pm 0.005$ | **$0.710 \pm 0.029$** | $1.908 \pm 0.043$ | **$1.920 \pm 0.075$** |
| | VGG-16 | 30.24M | BNNF | $0.576 \pm 0.001$ | **$0.775 \pm 0.009$** | $0.548 \pm 0.005$ | **$0.789 \pm 0.022$** | $1.540 \pm 0.018$ | **$1.549 \pm 0.090$** |
| | | | BNNR | $0.576 \pm 0.001$ | **$0.755 \pm 0.003$** | $0.548 \pm 0.005$ | **$0.718 \pm 0.005$** | $1.540 \pm 0.018$ | **$1.543 \pm 0.069$** |

the training distribution (e.g. MNIST, CIFAR-10), while the OoD dataset consists of samples that differ significantly from the training set (e.g. Fashion-MNIST for MNIST-trained models, SVHN or TinyImageNet for CIFAR-trained models). A model with strong epistemic uncertainty should assign higher entropy to OoD samples. Therefore, we analyse predictive entropy, calibration metrics, and OoD detection performance across both types of datasets.

**Uncertainty via Predictive Entropy and OoD Detection.** Our interval network outputs, for each input $x$, a pair of class–logit vectors $(\ell^{\mathrm{L}}, \ell^{\mathrm{U}}) \in \mathbb{R}^C$ representing lower and upper logits. We form a single logit vector by midpoint aggregation $\tilde{\ell} = \frac{1}{2}(\ell^{\mathrm{L}} + \ell^{\mathrm{U}})$, and convert to class probabilities with temperature $T$ (default $T=1$) via $p_T(y{=}c \mid x) = \mathrm{softmax}(\tilde{\ell}/T)_c$. We quantify predictive uncertainty using the (Shannon) predictive entropy $H_T(x) = -\sum_{c=1}^{C} p_T(y{=}c \mid x) \log p_T(y{=}c \mid x)$. For the out-of-distribution (OoD) set $\mathcal{D}_{\mathrm{ood}}$ we report the mean OoD entropy $\overline{H}_T^{\mathrm{OoD}} = |\mathcal{D}_{\mathrm{ood}}|^{-1} \sum_{x \in \mathcal{D}_{\mathrm{ood}}} H_T(x)$, and analogously compute the mean entropy on the in-distribution (ID) test set. We further assess OoD separability by using $H_T(x)$ as a scalar score (higher implies more OoD-like). Concatenating the ID and OoD scores with labels $\{0, 1\}$ (ID as 0, OoD as 1), we compute the area under the ROC curve (AUROC) and the area under the precision–recall curve (AUPRC). Thus, in our experiments *entropy is the uncertainty measure*: higher entropy indicates greater predictive uncertainty and is used directly to evaluate both calibration (through mean entropy and NLL/ECE) and OoD detection (via AUROC/AUPRC). Table 3 shows that Epi-Wrapper consistently improves OoD detection performance across all datasets and architectures. In particular, it achieves higher AUROC, AUPRC, and entropy scores compared to the INN baseline, demonstrating stronger separability between iD and OoD samples. This trend is further confirmed by the ROC and PRC curves in Figure 5a and Figure 5b,

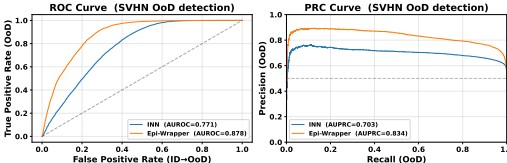 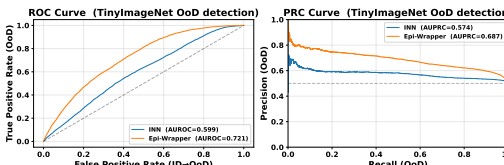

(a) ROC and PRC curves for iD CIFAR-10 vs. SVHN OoD detection using ResNet-18 with BNNR (Molchanov et al., 2017).

(b) ROC and PRC curves for iD CIFAR-100 vs. TinyImageNet OoD detection using VGG-16 with BNNR (Molchanov et al., 2017).

Figure 5: ROC and PRC results for OoD detection on SVHN and TinyImageNet benchmarks using INN and Epi-Wrapper. The curves illustrate performance trade-offs under entropy-based scoring, with AUROC and AUPRC values reported in the legends.

where Epi-Wrapper outperforms the INN baseline. Additional ROC/PRC results are presented in Section C.4, and calibration results (NLL/ECE) are reported in Appendix C.5.

### 4.4 ADDITIONAL RESULTS AND DETAILS IN APPENDIX

We summarize the contents of the Appendix in this section. Appendix A provides theoretical background concepts, including Belief functions, Dirichlet distributions, and the functionality of Interval Neural Networks (INNs). Appendix B contains implementation details, dataset descriptions, and Bayesian baseline setups. It also specifies the backbone architectures (MLP, LeNet-5, ResNet-18, and VGG-16), BNN training procedures, and computational costs. Appendix C presents additional ablation studies, including distributional choices over the simplex, the effect of budget set size, and the effect of the number of closed intervals. It also contains details on fine-tuning large-scale models, hyperparameter settings, ROC and PRC computation procedures, and results on calibration and likelihood evaluation. Appendix D discusses modeling epistemic uncertainty in parameter space versus target space. Appendix E outlines a preliminary idea for extending Epi-Wrapper to regression tasks.

## 5 CONCLUSION AND FUTURE DIRECTIONS

This paper introduces *Epistemic Wrapper*, a methodology that extends higher-order uncertainty representation into the parameter space of neural networks. Building on BNNs, it transforms their outputs into belief-function posteriors, enabling richer and more expressive quantification of epistemic uncertainty. The method is robust, efficient, and applicable across architectures, with experiments on four benchmark datasets showing consistent improvements in uncertainty estimation. A limitation is that Epi-Wrapper's performance depends on the quality of the underlying BNN, as poorly trained or high-variance models may yield degraded belief-function outputs. As future work, we aim to integrate Epistemic Wrapper with Bayesian Neural Operators for structured uncertainty quantification in function space, particularly for complex physical systems governed by PDEs. Another direction is to use the wrapped weights to construct predictive random sets in the target space, advancing reliable uncertainty-aware learning.

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

## A  THEORETICAL BACKGROUND CONCEPTS

### A.1  BELIEF FUNCTIONS

Belief functions Cuzzolin (2014b; 2018b), grounded in the mathematical framework of random sets, were initially introduced by Dempster (Dempster, 1967) and later formalized by Shafer (Shafer, 1976) as an alternative model for subjective belief to Bayesian probability. As mathematical objects, they have been extensively studied from an original geometric point of view Cuzzolin & Frezza

(2001); Cuzzolin (2003; 2004; 2008b; 2010c;b). Ways of transforming belief functions into Bayesian probabilities or possibility measures have also been investigated Cuzzolin (2009; 2010a; 2011; 2014a; July 2011; 2007).

In finite domains, such as a collection of classes, belief functions are characterised by a *basic probability assignment* (BPA) (Shafer, 1976), which is a set function $m : 2^{\Theta} \to [0, 1]$ satisfying $m(\emptyset) = 0$ and $\sum_{A \subseteq \Theta} m(A) = 1$. The value $m(A)$ is interpreted as the probability mass directly assigned to subset $A \subseteq \Theta$ in a random-set formulation (Smets, 1991). Subsets $A$ of $\Theta$ with $m(A) > 0$ are referred to as *focal elements*. Classical belief functions extend the notion of discrete mass functions by assigning normalized, non-negative mass values not only to elements $\theta \in \Theta$ but to subsets of $\Theta$, governed by:

$$m(A) \geq 0, \forall A \subseteq \Theta, \quad \sum_{A \subseteq \Theta} m(A) = 1. \tag{7}$$

The belief function $Bel(A)$ associated with a mass function $m$ is defined as the total mass assigned to all subsets $B \subseteq A$. Conversely, $m$ can be recovered from $Bel$ through Moebius inversion Shafer (1976):

$$Bel(A) = \sum_{B \subseteq A} m(B), \quad m(A) = \sum_{B \subseteq A} (-1)^{|A \setminus B|} Bel(B). \tag{8}$$

This formulation demonstrates that classical probability measures are a special case of belief functions, assigning mass exclusively to singletons.

## A.2 DIRICHLET DISTRIBUTION

The Dirichlet distribution is a continuous multivariate probability distribution defined over the $(K-1)$ simplex:

$$\Delta^{K-1} = \left\{ \boldsymbol{x} \in \mathbb{R}^K \ \middle| \ x_i \geq 0, \sum_{i=1}^{K} x_i = 1 \right\}.$$

It is parameterized by a concentration vector $\boldsymbol{\alpha} = (\alpha_1, \ldots, \alpha_K)$ with each $\alpha_i > 0$, and its probability density function is given by:

$$f(x_1, \ldots, x_K; \alpha_1, \ldots, \alpha_K) = \frac{1}{B(\boldsymbol{\alpha})} \prod_{i=1}^{K} x_i^{\alpha_i - 1}, \tag{9}$$

where $B(\boldsymbol{\alpha})$ is the multivariate Beta function:

$$B(\boldsymbol{\alpha}) = \frac{\prod_{i=1}^{K} \Gamma(\alpha_i)}{\Gamma\left(\sum_{i=1}^{K} \alpha_i\right)}.$$

The shape of the Dirichlet distribution is governed by the values of $\boldsymbol{\alpha}$. Higher values lead to more concentrated distributions around the center of the simplex, while lower values result in more dispersed or sparse distributions. Figure 6 illustrates how different $\boldsymbol{\alpha}$ values affect the distribution over the 2D simplex.

The Dirichlet distribution is widely used in Bayesian statistics, especially for modeling topics in documents and for representing uncertainty Gelman et al. (2013).

## A.3 INTERVAL NEURAL NETWORKS (INNs)

Traditional *interval neural networks* use deterministic interval-based inputs, outputs, and parameters (weights and biases) for each node. The forward propagation in the $l^{\text{th}}$ layer of INNs is expressed as:

$$\begin{aligned} [\underline{\boldsymbol{a}}, \overline{\boldsymbol{a}}]^l &= \sigma^l([\underline{\boldsymbol{\omega}}, \overline{\boldsymbol{\omega}}]^l \odot [\underline{\boldsymbol{a}}, \overline{\boldsymbol{a}}]^{l-1} \oplus [\underline{\boldsymbol{b}}, \overline{\boldsymbol{b}}]^l) \\ &= [\sigma^l(\underline{\boldsymbol{o}} + \underline{\boldsymbol{b}}), \sigma^l(\overline{\boldsymbol{o}} + \overline{\boldsymbol{b}})] \text{ with} \\ [\underline{\boldsymbol{o}}, \overline{\boldsymbol{o}}]^l &= [\underline{\boldsymbol{\omega}}, \overline{\boldsymbol{\omega}}]^l \odot [\underline{\boldsymbol{a}}, \overline{\boldsymbol{a}}]^{l-1}, \end{aligned} \tag{10}$$

where $\oplus$, $\ominus$, and $\odot$ represent interval addition, subtraction, and multiplication, respectively (Hickey et al., 2001). The terms $[\underline{\boldsymbol{a}}, \overline{\boldsymbol{a}}]^l$, $[\underline{\boldsymbol{a}}, \overline{\boldsymbol{a}}]^{l-1}$, $[\underline{\boldsymbol{\omega}}, \overline{\boldsymbol{\omega}}]^l$, and $[\underline{\boldsymbol{b}}, \overline{\boldsymbol{b}}]^l$ denote the interval-formed outputs of the

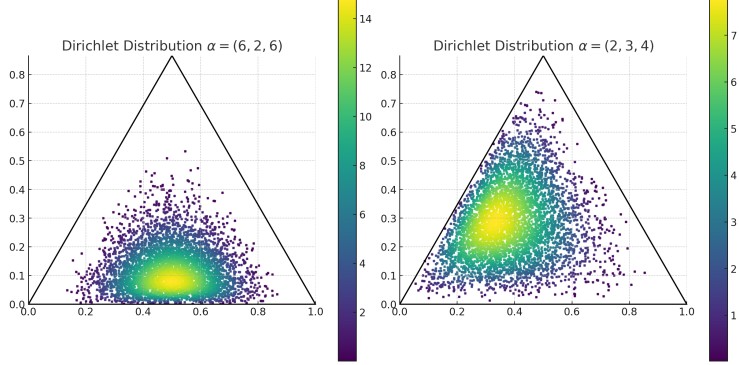

Figure 6: Probability densities of the Dirichlet distribution as functions on the 2D simplex: $\alpha =$ (6,2,6) (left), $\alpha =$ (2,3,4) (right).

$l^{th}$ and $(l-1)^{th}$ layers, as well as the intervals of weights and biases of the $l^{th}$ layer, respectively. $\sigma^l(\cdot)$ is the activation function of the $l^{th}$ layer, which must be monotonically increasing. The application of interval arithmetic (Hickey et al., 2001) in eq. equation 10 grants INNs the 'set constraint' property. Specifically, for any $\boldsymbol{a}^{l-1} \in [\underline{\boldsymbol{a}}, \overline{\boldsymbol{a}}]^{l-1}$, $\boldsymbol{\omega}^l \in [\underline{\boldsymbol{\omega}}, \overline{\boldsymbol{\omega}}]^l$, and $\boldsymbol{b}^l \in [\underline{\boldsymbol{b}}, \overline{\boldsymbol{b}}]^l$, the constraint in eq. equation 11 consistently holds.

$$\boldsymbol{a}^l = \sigma^l(\boldsymbol{\omega}^l \cdot \boldsymbol{a}^{l-1} + \boldsymbol{b}^l) \in [\underline{\boldsymbol{a}}, \overline{\boldsymbol{a}}]^l. \tag{11}$$

If $[\underline{\boldsymbol{a}}, \overline{\boldsymbol{a}}]$ is non-negative, such as the output of RELU activation, the calculation of $[\underline{\boldsymbol{o}}, \overline{\boldsymbol{o}}]$ in eq. equation 10 can be simplified as:

$$\begin{aligned} \underline{\boldsymbol{o}} &= \min\{\underline{\boldsymbol{\omega}}, \boldsymbol{0}\} \cdot \overline{\boldsymbol{a}} + \max\{\underline{\boldsymbol{\omega}}, \boldsymbol{0}\} \cdot \underline{\boldsymbol{a}} \\ \overline{\boldsymbol{o}} &= \max\{\overline{\boldsymbol{\omega}}, \boldsymbol{0}\} \cdot \overline{\boldsymbol{a}} + \min\{\overline{\boldsymbol{\omega}}, \boldsymbol{0}\} \cdot \underline{\boldsymbol{a}} \end{aligned} \tag{12}$$

### A.4 INTERVAL WEIGHTS FROM WRAPPED POSTERIORS

Dirichlet sampling yields probability vectors on the simplex, not intervals. In our framework, each Dirichlet sample is therefore *mapped* to an interval so that it can be propagated through the Interval Neural Network (INN). Given a Dirichlet sample $\boldsymbol{x} = (x_1, x_2, x_3)$, we define the corresponding interval weight as

$$\left[\underline{\boldsymbol{\omega}}, \overline{\boldsymbol{\omega}}\right] = \left[\min_k x_k, \ \max_k x_k\right].$$

This interval summarises the dispersion encoded by the Dirichlet posterior and fits naturally within the INN's interval-based forward rules (see Eq. equation 10). These are the *wrapped* weights produced by the Epistemic Wrapper.

For comparison, INNs using *unwrapped* BNN parameters construct interval weights from Gaussian posteriors via

$$\underline{\boldsymbol{\omega}} = \mu - \sigma, \qquad \overline{\boldsymbol{\omega}} = \mu + \sigma,$$

which yields a symmetric interval around the mean. Thus, during inference the INN operates with two types of interval weights: (i) Dirichlet-derived intervals for wrapped parameters, and (ii) Gaussian-derived intervals for unwrapped parameters. Aside from the initial interval construction, the forward propagation is identical.

It is important to emphasise that our use of the Dirichlet differs from its classical role in evidential deep learning. There, Dirichlet samples are treated as categorical probability vectors. In our case, the Dirichlet is fitted to mass vectors defined over closed parameter intervals obtained via Möbius inversion. Its domain is therefore a mass function over sets of parameter values, not a categorical sample space. The interval mapping reflects this domain shift: the Dirichlet captures uncertainty over parameter intervals, and each sample yields an interval bound suitable for stable propagation through the INN.

---

**Algorithm 1** Epistemic Wrapper for Parameter-Space Epistemic Uncertainty

---

**Require:** Pretrained BNN with variational posterior $q(\omega)$; posterior samples $\{\omega^{(s)}\}_{s=1}^{S}$; number of closed intervals $M$; budgeting percentage $\beta$.

1: **EpistemicWrapper**$(q(\omega), \{\omega^{(s)}\})$
2: Select a subset of weights $\mathcal{W}$ using the budgeting criterion (e.g., high-mean or random)
3: **for** each weight $\omega \in \mathcal{W}$ **do**
4:     **(1) Estimate posterior parameters**
5:     Compute posterior mean $\mu$ and standard deviation $\sigma$
6:     **(2) Dynamic truncation**
7:     mult $\leftarrow \min(5.0, \ 1/\sigma)$
8:     $a_{\min} = \mu - \text{mult} \cdot \sigma$
9:     $a_{\max} = \mu + \text{mult} \cdot \sigma$
10:     **(3) Construct closed-interval grid**
11:     Discretize $[a_{\min}, a_{\max}]$ into $M$ closed intervals $\{A_i = [a_i, b_i]\}_{i=1}^{M}$
12:     **(4) Compute belief and plausibility**
13:     **for** each interval $A_i$ **do**
14:         $Pl(A_i) \leftarrow \sup_{\omega \in A_i} q(\omega)$
15:         $Bel(A_i) \leftarrow 1 - Pl(A_i^c)$
16:     **end for**
17:     **(5) Möbius inversion (Belief $\rightarrow$ Mass)**
18:     **for** each interval $A_i$ **do**
19:         $m(A_i) = \sum_{B \subseteq A_i} (-1)^{|A_i \setminus B|} Bel(B)$
20:     **end for**
21:     **(6) Normalize mass values**
22:     $\tilde{m}_i = m(A_i) / \sum_{j=1}^{M} m(A_j)$
23:     **(7) Project onto 3D simplex**
24:     Map $\{\tilde{m}_i\}_{i=1}^{M}$ to a 3D simplex via barycentric projection
25:     **(8) Compute L-moments**
26:     Compute weighted first L-moment $L_1$ and second L-moment $L_2$
27:     **(9) Fit Dirichlet distribution**
28:     $\alpha_k = L_{1,k} \left( \frac{L_{1,k}(1 - L_{1,k})}{L_{2,k}} - 1 \right), \ \ k = 1, 2, 3$
29:     Clamp $\alpha_k$ to ensure positivity
30:     Store $\boldsymbol{\alpha}_\omega = (\alpha_1, \alpha_2, \alpha_3)$
31: **end for**
32: **return** $\{\text{Dir}(\boldsymbol{\alpha}_\omega)\}_{\omega \in \mathcal{W}}$

---

## B  EXPERIMENTAL SETUP

### B.1  IMPLEMENTATION DETAILS

All experiments are implemented using the TensorFlow framework (version 2.13.1), with probabilistic modeling and inference carried out using TensorFlow Probability (TFP) (version 0.21.0). TFP is a library for statistical analysis and probabilistic reasoning that integrates seamlessly with TensorFlow. We employed two Bayesian baseline models: 'BNNR', which uses Auto-Encoding Variational Bayes (Kingma & Welling, 2013) with the local re-parameterization trick (Molchanov et al., 2017), and 'BNNF', which leverages the Flipout gradient estimator along with a negative evidence lower bound (ELBO) loss (Wen et al., 2018). Both baselines are implemented using TFP's built-in variational layers and loss functions. Our complete pipeline, including training, inference, and evaluation of the Epistemic Wrapper is built entirely in TensorFlow 2.13.1 and executed on a machine equipped with 8× NVIDIA A30 GPUs. For clarity, a full algorithmic description of the Epistemic Wrapper is provided in Algorithm 1.

## B.2 DATASETS

We evaluated the performance of the Epistemic Wrapper on four classification benchmarks: MNIST (LeCun, 1998), Fashion-MNIST Xiao et al. (2017), CIFAR-10 (Krizhevsky, 2009a) and CIFAR-100 Krizhevsky (2009b). Following are the details of the iD and OoD datasets.

### B.2.1 ID DATASETS

**MNIST** dataset comprises 70,000 greyscale images of handwritten digits (0-9), each with a resolution of $28 \times 28$ pixels, and is mostly used for classification and pattern recognition tasks due to its simplicity and accessibility.

**Fashion MNIST** serves as a more challenging alternative to MNIST, containing 70,000 greyscale images of fashion items, such as shirts, shoes, and bags, also at same resolution of $28 \times 28$ pixels. This dataset provides a greater diversity in texture and structure, making it suitable for evaluating model's generalization capabilities.

**CIFAR-10** is a collection of 60,000 color images (split into 50,000 training and 10,000 testing samples) across 10 classes, including animals and vehicles, with each image having a resolution of $32 \times 32$ pixels. CIFAR-10 is particularly valuable for assessing models in tasks involving color and more complex spatial patterns.

**CIFAR-100** consists of 60,000 color images, each of size $32 \times 32$ pixels with three RGB channels, divided into 50,000 training images and 10,000 test images. The dataset contains 100 fine-grained classes, with each class having 600 samples, making it a more challenging extension of the CIFAR-10 dataset. Unlike CIFAR-10, which includes only 10 broad categories, CIFAR-100 introduces a hierarchical structure, grouping its 100 classes into 20 superclasses based on semantic similarity.

### B.2.2 OOD DATASETS

**SVHN** (Street View House Numbers) is a real-world image dataset obtained from house numbers captured by Google Street View. It consists of over 600,000 digit images (0–9), cropped from street number plates, with each image being $32 \times 32$ pixels in RGB format. Unlike the balanced and object-centric nature of CIFAR datasets, SVHN exhibits high variability in illumination, orientation, and background clutter. It is primarily designed for digit recognition tasks but is widely adopted as an OoD dataset when models are trained on natural object datasets such as CIFAR-10 or CIFAR-100, due to its distinct visual domain.

**TinyImageNet** is a subset of the ImageNet dataset constructed for benchmarking under constrained input dimensions. It contains 200 object classes with 500 training images, 50 validation images, and 50 test images per class, leading to a total of 100,000 images. Each image is downsampled to $64 \times 64$ pixels, significantly smaller than the original ImageNet resolution. The dataset retains substantial intra-class variation and fine-grained categories. Due to its larger diversity and semantic distance from CIFAR classes, TinyImageNet serves as a strong OoD benchmark for evaluating model robustness and generalization.

## B.3 BAYESIAN BASELINES

In our current implementation, we employ standard and widely-used VI methods such as Auto-Encoding Variational Bayes with local reparameterization BNNR (Auto-Encoding Variational Bayes (Kingma & Welling, 2013) with the local re-parameterization trick (Molchanov et al., 2017)), and BNNF (Flipout gradient estimator with the negative evidence lower bound loss (Wen et al., 2018)). These provide reliable and efficient posterior approximations and are well-supported in TensorFlow, offering a stable foundation for Bayesian neural network training. The core design of the Epistemic Wrapper is independent of the specific VI family used to approximate the posterior. It operates on sampled posterior weights, and thus can naturally extend to richer VI families. We chose mean-field-based VI families in this initial study for their computational tractability and widespread adoption.

### B.4 BACKBONES

We utilized four backbone models in our experiments: MLP, LeNet-5 LeCun et al. (1998), ResNet-18 (He et al., 2016), and VGG-16 Simonyan & Zisserman (2015). The architectural details of each backbone are provided below.

**MLP** is composed of an input layer, a single hidden layer and an output layer. The input layer processes the input data with a shape that corresponds to the dimensions of the dataset. For grayscale datasets (MNIST and Fashion MNIST), the input shape is $28 \times 28 \times 1$, and for CIFAR-10 and CIFAR-100, the input shape is $32 \times 32 \times 3$. A flattening layer flattens the input into a single-dimensional vector to be fed to the subsequent dense layers. 'DenseFlipout Layers' are implemented using TFP. They approximate the weight posterior distributions using a Flipout Monte Carlo estimator, which reduces the variance of gradient estimates during backpropagation. The first dense layer contains hidden units with ReLU activation, followed by a dropout layer to prevent overfitting. The second dense layer, which acts as the output layer, maps to the number of classes in the dataset.

**LeNet-5** architecture is adapted into a fully Bayesian framework using variational inference with Flipout layers from TensorFlow Probability. Both convolutional and fully connected layers are replaced with their Flipout-based counterparts. The input shape is dataset dependent: $28 \times 28 \times 1$ for grayscale datasets such as MNIST and Fashion-MNIST, and $32 \times 32 \times 3$ for RGB datasets such as CIFAR-10 and CIFAR-100. The model begins with a 'Convolution2DFlipout' layer with 6 filters and a $5 \times 5$ kernel, followed by an average pooling layer. This is followed by a second 'Convolution2DFlipout' layer with 16 filters and another $5 \times 5$ kernel, again followed by average pooling. The output is then flattened and passed through two fully connected variational layers: a 'DenseFlipout' layer with 120 units and ReLU activation, followed by a second 'DenseFlipout' layer with 84 units. The final classification is performed by a 'DenseFlipout' layer with softmax activation and a number of units equal to the number of classes.

**ResNet-18** model leverages Bayesian Convolutional Neural Networks (Bayesian CNNs) with Flipout and Reparameterization layers from TensorFlow Probability, enabling weight uncertainty modeling. The architecture consists of four main residual blocks, with convolutional layers followed by batch normalization and ReLU activation. The convolutional layers employ Bayesian weight posterior distributions, where the kernel weights follow a Gaussian posterior parameterized by mean and variance. These distributions are constrained using a log-variance regularization technique, ensuring numerical stability. The weight posteriors are sampled using the Mean-Field Variational Inference approach, enabling Bayesian updates during training. The ResNet-18 backbone begins with an initial convolutional layer followed by four residual blocks, each progressively increasing the number of filters from 64 to 512. The residual connections allow gradient flow through the network, ensuring stable training. The final layers include average pooling, flattening, and a fully connected Bayesian dense layer with Flipout, producing the classification logits.

**VGG-16** model integrates Bayesian inference into the classical VGG-16 architecture to enable principled uncertainty estimation in deep learning. The standard convolutional layers are replaced with Bayesian Convolutional Neural Networks (Bayesian CNNs) using Convolution2DReparameterization and Convolution2DFlipout layers from TensorFlow Probability. These layers approximate posterior distributions over weights using Mean-Field Variational Inference, ensuring reliable uncertainty quantification. VGG-16 follows a deep convolutional architecture with 16 layers, consisting of multiple stacked convolutional layers with small $3 \times 3$ filters, followed by max pooling layers to progressively reduce spatial dimensions. The Bayesian adaptation maintains this structure while introducing posterior weight sampling in convolutional layers, ensuring that the feature extraction process incorporates uncertainty information. Batch normalization and ReLU activation are applied to enhance convergence stability, while Bayesian priors constrain weight posteriors, preventing overconfidence in predictions. The final classification layers include Bayesian fully connected layers with Flipout, which sample weights during inference to produce uncertainty-aware predictions.

**BNNR vs. BNNF instantiations:** Both BNNR and BNNF share the same Bayesian architectures of all above mentioned backbones but differ in their variational inference strategies:

**BNNR** (Bayesian Neural Network with Reparameterization) uses the Auto-encoding Variational Bayes Kingma & Welling (2013) along with the local reparameterization trick Molchanov et al.

(2017). This variant estimates the evidence lower bound (ELBO) and applies Gaussian posterior sampling locally per activation.

**BNNF** (Bayesian Neural Network with Flipout) uses the Flipout estimator Wen et al. (2018), which decorrelates the gradient estimates across examples in a mini-batch, leading to lower variance during optimization. BNNF directly applies Flipout-based sampling in both convolutional and dense layers and uses a negative ELBO as the loss function.

These two baselines allow us to evaluate the effectiveness of our Epistemic Wrapper under different stochastic inference regimes.

### B.5 TRAINING DETAILS

As a baseline, we use standard variational Bayesian Neural Networks (BNNs) (Blei et al., 2017), starting with a classical Multilayer Perceptron (MLP) architecture. The Bayesian MLP is trained using the Evidence Lower Bound (ELBO) objective, which combines the negative log-likelihood (NLL) with a Kullback-Leibler (KL) divergence regularization term. The NLL is computed via softmax cross-entropy, and the KL divergence measures the distance between the approximate posterior and the prior distributions over the weights. The MLP models are trained for 20 epochs on MNIST and Fashion-MNIST using the Adam optimizer. The Bayesian LeNet-5 is trained under both BNNR and BNNF setups using the same ELBO-based training strategy. Models are trained for 500 epochs on CIFAR-10 with Adam optimizer, batch size 32. We apply standard data augmentation techniques (random horizontal flipping and cropping). For deeper architectures such ResNet-18 and VGG-16, we adopt dataset-specific training schedules. Models trained on CIFAR-10 are trained for 50 epochs, while those trained on CIFAR-100 are trained for 200 epochs. Batch normalization is applied after each convolutional layer. Both BNNR and BNNF variants are implemented using Flipout-compatible variational layers from TensorFlow Probability and are trained from scratch on a single NVIDIA A30 GPU.

### B.6 COMPUTATIONAL COST

The computational overhead introduced by the Epistemic Wrapper is modest. The wrapper operates only on a small, selected subset of weights (5% for MLPs and 0.1% for large-scale models), chosen via our posterior-based budgeting criterion. Since the procedure is applied post hoc to sampled BNN weights, it does not interfere with the forward or backward passes of the underlying Bayesian network. Training cost therefore remains unchanged, and the additional runtime appears only during the wrapping stage. **Hardware:** All experiments were conducted on a workstation equipped with CPUs (32 cores, 2.8 GHz) and 8× NVIDIA A30 GPUs (24 GB each). Each experiment was executed on a single GPU. Table 4 reports the total wrapping time for each model–dataset combination. For instance, applying the wrapper to CIFAR-10 with ResNet-18 takes 403-406 seconds on a single NVIDIA A30 GPU. The overhead scales primarily with model size, as expected, but remains practical even for large architectures such as VGG-16. The wrapper therefore maintains compatibility with large-scale BNNs without imposing substantial computational cost.

## C EXPERIMENTAL RESULTS

### C.1 ADDITIONAL ABLATION STUDIES

**Distributional Choices over the Simplex:** To assess the modelling choice of using a Dirichlet distribution for belief posteriors representation, we perform an ablation study comparing it to two alternative probability distributions defined over the simplex. The following experiments compare the Dirichlet with a Generalized Dirichlet and a Beta stick-breaking model, evaluating their ability to capture the structure of the normalized masses in the proposed methodology. While any valid Probability Density Function (PDF) defined over the simplex could, in principle, be used to model belief distributions, Dirichlet distributions have consistently demonstrated empirical effectiveness for representing epistemic uncertainty in neural networks Stirn et al. (2019). To assess this modelling choice more critically, we conduct an ablation study comparing the use of three distributions defined on the probability simplex: (i) a standard Dirichlet distribution fitted using the method of L-moments, (ii) a Generalized Dirichlet distribution, and (iii) a Beta stick-breaking model. The Dirichlet distribu-

Table 4: End-to-end runtime of the Epistemic Wrapper on a single NVIDIA A30 GPU. Times correspond to wrapping the selected subset of posterior weights (5% for MLPs and 0.1% for LeNet-5, ResNet-18 and VGG-16). Training time of the underlying BNN is unchanged.

| DATASET | BACKBONE | # PARAMS | BASELINE | BUDGETING % | COMPUTATIONAL TIME (SECONDS) |
|---|---|---|---|---|---|
| MNIST | MLP | 12.7K | BNNF | 5.0% | 17.23 |
| FASHION-MNIST | MLP | 12.7K | BNNF | 5.0% | 17.07 |
| CIFAR-10 | LENET-5 | 166.3K | BNNF | 0.1% | 8.65 |
| | | | BNNR | 0.1% | 9.33 |
| | RESNET-18 | 9.82M | BNNF | 0.1% | 403.56 |
| | | | BNNR | 0.1% | 405.53 |
| | VGG-16 | 30.24M | BNNF | 0.1% | 4003.19 |
| | | | BNNR | 0.1% | 4160.05 |
| CIFAR-100 | RESNET-18 | 9.8M | BNNF | 0.1% | 1304.67 |
| | | | BNNR | 0.1% | 1317.49 |
| | VGG-16 | 30.24M | BNNF | 0.1% | 4066.00 |
| | | | BNNR | 0.1% | 4180.30 |

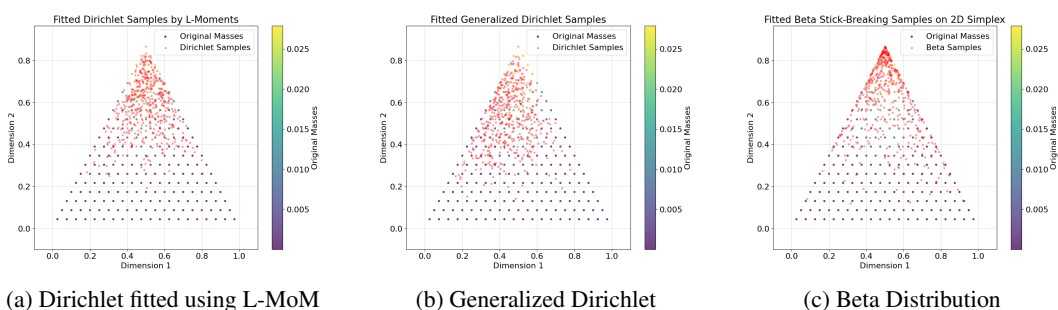

(a) Dirichlet fitted using L-MoM     (b) Generalized Dirichlet     (c) Beta Distribution

Figure 7: Distributional Choices over the Simplex

tion is used as our baseline because it offers a mathematically simple, symmetric, and interpretable formulation that is widely adopted in evidential deep learning literature. It allows us to encode a single mode of belief mass and is straightforward to parameterize using closed-form moments. The Generalized Dirichlet distribution extends this by introducing additional flexibility via a second set of parameters, enabling skewness and more complex belief shapes. Finally, the Beta stick-breaking model introduces an alternative constructive approach by allocating belief mass sequentially from Beta-distributed proportions, resulting in a valid but directionally expressive distribution over the simplex Stirn et al. (2019). All three models are fitted to the same belief mass vectors, and their sample distributions are visualized in Figures 7a, 7b, and 7c. While the Dirichlet and Generalized Dirichlet distributions align well with the center of mass observed in the belief functions, the Beta stick-breaking model demonstrates an equally valid fit but with a distinct shape and construction. This ablation supports the rationale that, while various PDFs on the simplex are valid for representing belief functions, the Dirichlet distribution remains a principled choice due to its mathematical simplicity, interpretability and established use in modelling epistemic uncertainty in neural networks.

**Effect of Budgeting Set Size:** Table 5 presents the classification accuracy of the Epistemic Wrapper on MNIST across varying budgeting percentages, both before and after fine-tuning. The percentage indicates the fraction of posterior weights selected for wrapping based on a high-mean criterion, while the number of intervals was fixed at 30. Before fine-tuning, we observe that moderate budgeting (e.g., 10% or 20%) results in significantly improved performance over the baseline INN, which achieved only 9.33% accuracy. In particular, wrapping only 10% of the weights led to a substantial increase in accuracy (45.34%), indicating that even a small subset of informative weights contributes meaningfully to uncertainty-aware decision-making. However, performance declines when budgeting exceeds 20%, which may be attributed to the limited capacity of the small-scale MLP model (with only 8 hidden units). In such a constrained architecture, wrapping a larger fraction of weights may reduce generalization by overfitting or introducing excessive variance in the wrapped ensemble.

Table 5: Performance comparison on MNIST across different budgeting percentages before and after fine-tuning.

| STRATEGY | INN ACCURACY (%) | EPI-WRAPPER BUDGETING PERCENTAGE (SELECTED WEIGHTS) | | | | | |
|---|---|---|---|---|---|---|---|
| | | 5% | 10% | 20% | 30% | 40% | 50% |
| BEFORE FINE-TUNING | $9.33 \pm 0.54$ | $25.46 \pm 1.57$ | $\mathbf{45.34 \pm 1.38}$ | $42.25 \pm 1.26$ | $32.62 \pm 2.00$ | $19.07 \pm 0.80$ | $14.08 \pm 0.94$ |
| AFTER FINE-TUNING | $91.12 \pm 0.08$ | $91.08 \pm 0.09$ | $91.83 \pm 0.04$ | $91.84 \pm 0.04$ | $\mathbf{91.85 \pm 0.13}$ | $91.82 \pm 0.09$ | $91.50 \pm 0.05$ |

After fine-tuning, performance improves and stabilizes across all budgeting levels. The accuracy remains consistently above 91% even with minimal weight wrapping, indicating that fine-tuning effectively adjusts the selected wrapped weights to better align with the underlying predictive task. The best performance (91.85%) is achieved at 30% budgeting, but all configurations from 10% to 50% perform comparably well, highlighting the robustness of the approach after refinement. These results confirm that wrapping a small subset of epistemically informative weights can significantly enhance predictive performance.

**Effect of Number of Closed Intervals:** To assess the impact of closed intervals for computing belief values leading to fitting a Dirichlet distribution over grid of mass values (as explained in sections 3.3 and 3.4). We conducted an ablation study by varying the number of intervals before fitting the Dirichlet distribution. Table 6 and Fig. 8 show the resulting $\alpha$ values computed using the method of *L-moments* for different numbers of intervals, with a fixed sample size of 5000. We observe that the estimated parameters stabilize around 30 intervals, beyond which changes become marginal. Based on this observation, we fix the number of intervals to 30 for all subsequent experiments to balance estimation stability and computational efficiency.

All ablation experiments are conducted on the MNIST dataset using a BNN MLP (hidden units= 8, samples =5000).

Table 6: Ablation study of Dirichlet $\alpha$ estimates using method of L-moment across varying numbers of intervals. Results are computed on 5000 samples.

| Number of Intervals | Estimated $\alpha$ Values |
|---|---|
| 10 | [1.0657, 4.5643, 1.0626] |
| 20 | [1.5958, 5.7176, 1.4097] |
| 30 | [1.6651, 6.1145, 1.6197] |
| 40 | [1.5272, 6.2213, 1.5665] |
| 50 | [1.6660, 6.3547, 1.6410] |
| 60 | [1.6935, 6.4927, 1.6669] |

## C.2 FINE-TUNING USING WRAPPER WEIGHTS

Fine-tuning is an essential stage of the Epistemic Wrapper pipeline. Once the second-order representation is constructed and the INN is initialised with Dirichlet-derived (wrapped) and Gaussian-derived (unwrapped) interval weights, the network must be adapted to the classification task under this new parameterisation. Fine-tuning restores discriminative performance while preserving the epistemic structure introduced by the Wrapper. The purpose of our method is therefore improved epistemic-aleatoric separation, calibration, interpretability, and OOD robustness-not state-of-the-art accuracy.

During inference, the INNs operate with two types of weights (as explained in section 3.5): 'unwrapped' weights, directly sampled from Gaussian posteriors, and 'wrapped' weights that are Dirichlet-derived intervals. Specifically, for a given unwrapped weight with mean $\mu$ and standard deviation $\sigma$, we define the interval as:

$$\text{Lower Bound} = \mu - k \cdot \sigma, \qquad \text{Upper Bound} = \mu + k \cdot \sigma,$$

where $k$ is a learnable, layer-specific scaling factor.

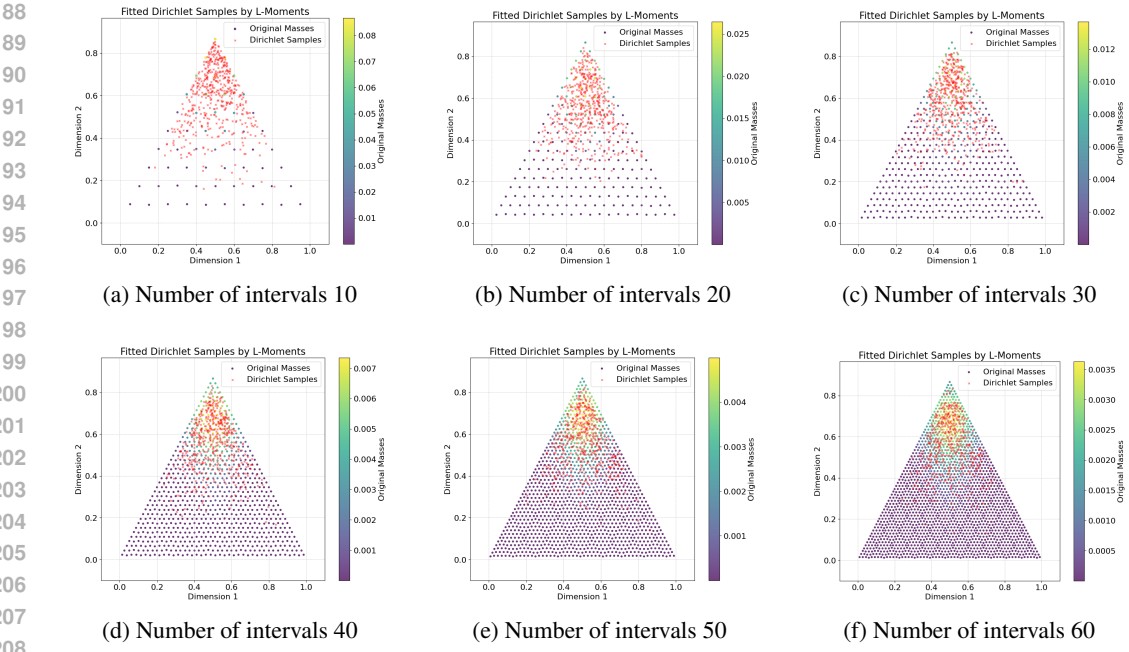

(a) Number of intervals 10          (b) Number of intervals 20          (c) Number of intervals 30

(d) Number of intervals 40          (e) Number of intervals 50          (f) Number of intervals 60

Figure 8: Varying Number of closed intervals

This scaling factor $k$ serves a similar purpose to the *tempering parameter* $\tau$ in variational Bayesian inference Osawa et al. (2019), which modulates the influence of the likelihood in the posterior. Smaller values of $k$ lead to narrower (sharper) intervals, corresponding to more confident parameter estimates; larger values encourage wider bounds and more conservative uncertainty. Unlike fixed-scale approaches (e.g., $\mu \pm \sigma$), our model learns $k$ jointly with the rest of the parameters during training. For numerical stability and positivity, we parameterize $k$ via a softplus transformation: $k = \log(1 + \exp(k_{\mathrm{raw}}))$, where $k_{\mathrm{raw}}$ is a trainable tensor. This uncertainty calibration mechanism allows each layer to adaptively control the initial spread of its weights, improving both optimization stability and predictive robustness. This is especially important in deep architectures, where poorly controlled interval propagation may lead to unstable gradients or diluted information. At inference time, our method uses these calibrated intervals to propagate uncertainty. In contrast, the baseline INN applies standard random initialization Kim (1993). Both models undergo fine-tuning, but only the Epi-Wrapper benefits from interval-aware initialization based on transformed posteriors.

### C.3   HYPERPARAMETER SETTINGS

The experimental setup of the main manuscript contains fixed hyperparameters such as number of closed intervals (30) and samples drawn from posterior distributions (5000). The budgeting strategy is applied consistently across experiments, with 5% of weights selected for small-scale model such as MLP, and 0.1% for larger architectures including LeNet-5, ResNet-18, and VGG-16.

### C.4   ROC AND PRC COMPUTATION FOR OoD DETECTION

**Scores and Labels.**    Given in-distribution (ID) samples with scores $\{s_i^{\mathrm{id}}\}_{i=1}^{n_{\mathrm{id}}}$ and out-of-distribution (OoD) samples with scores $\{s_j^{\mathrm{ood}}\}_{j=1}^{n_{\mathrm{ood}}}$, we form

$$\mathbf{s} = \big[s_1^{\mathrm{id}}, \ldots, s_{n_{\mathrm{id}}}^{\mathrm{id}},\ s_1^{\mathrm{ood}}, \ldots, s_{n_{\mathrm{ood}}}^{\mathrm{ood}}\big], \qquad \mathbf{y} = \big[0, \ldots, 0,\ 1, \ldots, 1\big],$$

where $y = 1$ denotes OoD and $y = 0$ denotes ID. In your code, each score is the predictive entropy

$$H(p) = -\sum_{c=1}^{C} p_c \log p_c,$$

computed from midpoint logits (averaging two heads) followed by softmax.

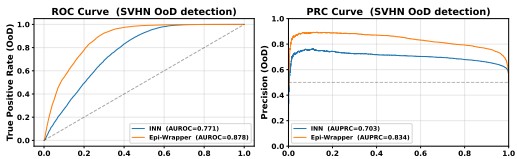

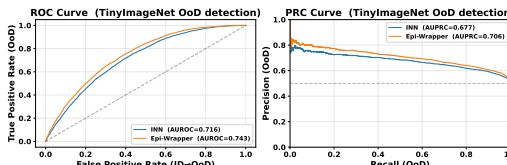

(a) ROC and PRC curves for iD CIFAR-10 vs. SVHN OoD detection using ResNet-18 with BNNR (Molchanov et al., 2017).

(b) ROC and PRC curves for iD CIFAR-100 vs. TinyImageNet OoD detection using ResNet-18 with BNNR (Molchanov et al., 2017).

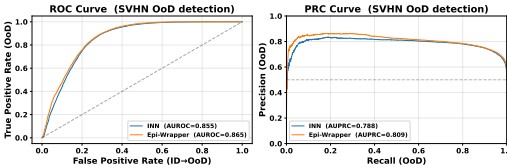

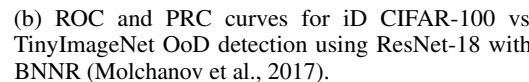

(c) ROC and PRC curves for iD CIFAR-10 vs. SVHN OoD detection using VGG-16 with BNNR (Molchanov et al., 2017).

(d) ROC and PRC curves for iD CIFAR-100 vs. TinyImageNet OoD detection using VGG-16 with BNNF (Wen et al., 2018).

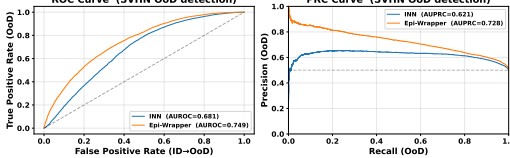

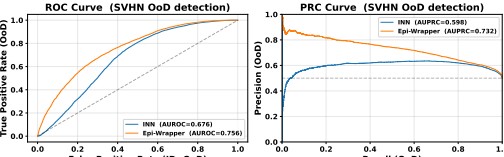

(e) ROC and PRC curves for iD CIFAR-10 vs. SVHN OoD detection using LeNet-5 with BNNR (Molchanov et al., 2017).

(f) ROC and PRC curves for iD CIFAR-100 vs. TinyImageNet OoD detection using LeNet-5 with BNNF (Wen et al., 2018).

Figure 9: ROC and PRC results for OoD detection on SVHN dataset using INN and Epi-Wrapper. The curves illustrate performance trade-offs under entropy-based scoring, with AUROC and AUPRC values reported in the legends.

**ROC:** For a threshold $\tau$, predict $\hat{y} = \mathbb{I}[s \geq \tau]$. Define

$$\text{TPR}(\tau) = \frac{\text{TP}(\tau)}{\text{TP}(\tau) + \text{FN}(\tau)}, \qquad \text{FPR}(\tau) = \frac{\text{FP}(\tau)}{\text{FP}(\tau) + \text{TN}(\tau)}.$$

Sweeping $\tau$ from $+\infty$ to $-\infty$ traces the ROC curve $\big(\text{FPR}(\tau), \text{TPR}(\tau)\big)$. The area under the ROC (AUROC) is computed by the trapezoidal rule over the curve.

**PRC:** Precision–Recall uses

$$\text{Precision}(\tau) = \frac{\text{TP}(\tau)}{\text{TP}(\tau) + \text{FP}(\tau)}, \qquad \text{Recall}(\tau) = \text{TPR}(\tau).$$

Sweeping $\tau$ yields the PRC $\big(\text{Recall}(\tau), \text{Precision}(\tau)\big)$. The area under the PRC (AUPRC) can be computed via the average-precision estimator or trapezoidal integration. A reference baseline is the positive prior $\pi = \frac{n_{\text{ood}}}{n_{\text{id}} + n_{\text{ood}}}$.

**Comparative Analysis.** The ROC and PRC plots in Figure 9 further highlight the performance differences between INN and Epi-Wrapper on OoD detection with SVHN as the outlier dataset. Across both backbone settings (ResNet-18 with BNNF and VGG-16 with BNNR), the Epi-Wrapper curves consistently dominate the INN baseline, reflecting higher true positive rates at lower false positive rates in ROC space, as well as improved precision across recall levels in PRC space. These visual trends align with the quantitative results reported in Table 3, confirming that parameter-space uncertainty modeling via Epi-Wrapper yields superior separability between in-distribution and out-of-distribution samples. The improvement is particularly pronounced in the PRC, where Epi-Wrapper maintains high precision even at challenging recall levels, suggesting its robustness under class-imbalance conditions common in OoD detection tasks.

Table 7: ECE and NLL for INN (baseline) vs. Epi-Wrapper (ours) across MNIST, Fashion-MNIST, CIFAR-10, and CIFAR-100 datasets using BNNF and BNNR baselines with MLP, LeNet-5, ResNet-18, and VGG-16 backbones. (Classification accuracy results are reported separately in Table 2.)

| DATASET | BACKBONE | # PARAMS | BASELINE | ECE ($\downarrow$) | | NLL ($\downarrow$) | |
| --- | --- | --- | --- | --- | --- | --- | --- |
| | | | | INN | EPI-WRAPPER | INN | EPI-WRAPPER |
| MNIST | MLP | 12.7K | BNNF | $0.011 \pm 0.031$ | $\mathbf{0.009 \pm 0.018}$ | $0.388 \pm 0.007$ | $\mathbf{0.298 \pm 0.004}$ |
| FASHION MNIST | MLP | 12.7K | BNNF | $0.035 \pm 0.012$ | $\mathbf{0.006 \pm 0.090}$ | $0.410 \pm 0.018$ | $\mathbf{0.332 \pm 0.019}$ |
| CIFAR-10 | LENET-5 | 166.3K | BNNF | $0.060 \pm 0.061$ | $\mathbf{0.059 \pm 0.019}$ | $1.365 \pm 0.090$ | $\mathbf{1.210 \pm 0.045}$ |
| | | | BNNR | $0.060 \pm 0.061$ | $\mathbf{0.053 \pm 0.006}$ | $\mathbf{1.365 \pm 0.090}$ | $1.608 \pm 0.009$ |
| | RESNET-18 | 9.82M | BNNF | $0.084 \pm 0.015$ | $\mathbf{0.041 \pm 0.013}$ | $0.616 \pm 0.070$ | $\mathbf{0.333 \pm 0.012}$ |
| | | | BNNR | $0.084 \pm 0.015$ | $\mathbf{0.053 \pm 0.012}$ | $0.616 \pm 0.070$ | $\mathbf{0.422 \pm 0.057}$ |
| | VGG-16 | 30.24M | BNNF | $0.073 \pm 0.012$ | $\mathbf{0.069 \pm 0.087}$ | $0.520 \pm 0.081$ | $\mathbf{0.451 \pm 0.033}$ |
| | | | BNNR | $0.073 \pm 0.012$ | $\mathbf{0.071 \pm 0.010}$ | $0.520 \pm 0.081$ | $\mathbf{0.462 \pm 0.054}$ |
| CIFAR-100 | RESNET-18 | 9.8M | BNNF | $0.059 \pm 0.049$ | $\mathbf{0.055 \pm 0.087}$ | $1.460 \pm 0.084$ | $\mathbf{1.451 \pm 0.088}$ |
| | | | BNNR | $0.059 \pm 0.049$ | $\mathbf{0.051 \pm 0.056}$ | $1.460 \pm 0.084$ | $\mathbf{1.437 \pm 0.010}$ |
| | VGG-16 | 30.24M | BNNF | $0.043 \pm 0.023$ | $\mathbf{0.040 \pm 0.013}$ | $2.141 \pm 0.089$ | $\mathbf{1.999 \pm 0.071}$ |
| | | | BNNR | $0.043 \pm 0.023$ | $\mathbf{0.039 \pm 0.045}$ | $2.141 \pm 0.089$ | $\mathbf{1.766 \pm 0.059}$ |

### C.5 CALIBRATION AND LIKELIHOOD EVALUATION

To evaluate the probabilistic calibration of our models, we compute two key metrics: 'Expected Calibration Error (ECE)' and 'Negative Log Likelihood (NLL)'. These metrics assess the alignment between predicted confidence and actual correctness (ECE), and the quality of probabilistic predictions (NLL).

**Expected Calibration Error (ECE):** quantifies the average discrepancy between the predicted confidence of a model and the observed precision. Predictions are binned into $M$ intervals (we use $M = 15$). For each bin, the average confidence and accuracy are computed, where $\text{conf}(B_m)$ is the average predicted maximum probability of samples in bin $B_m$. The ECE is the weighted average of the absolute difference between these values across all bins:

$$\text{ECE} = \sum_{m=1}^{M} \frac{|B_m|}{n} \left| \text{acc}(B_m) - \text{conf}(B_m) \right|, \tag{13}$$

where $n$ is the total number of samples, and $B_m$ denotes the $m$-th bin.

**Negative Log Likelihood (NLL):** captures how well a model's predicted probabilities match the true labels. It penalizes incorrect predictions with high confidence and rewards well-calibrated probability distributions:

$$\text{NLL} = -\frac{1}{n} \sum_{i=1}^{n} \log \hat{p}_{i,y_i}, \tag{14}$$

where $\hat{p}_{i,y_i}$ is the predicted probability for the correct class.

Table 7 summarizes the performance of the baseline INN and the proposed Epi-Wrapper model on the four datasets. Across almost all settings, Epi-Wrapper achieves lower ECE and NLL compared to the INN baseline, highlighting its superior probabilistic calibration and confidence estimation. These improvements suggest that, beyond classification accuracy, Epi-Wrapper provides more trustworthy uncertainty estimates, which is especially important for decision-critical applications where model confidence is as crucial as prediction correctness.

## D MODELING EPISTEMIC UNCERTAINTY: PARAMETER VS. TARGET SPACE

Epistemic uncertainty (EU) originates from limited knowledge about the model parameters themselves, a view grounded in Bayesian learning where posterior distributions over weights directly encode parameter uncertainty. Although several recent approaches such as Evidential Deep Learning (EDL) (Sensoy et al., 2018) and credal set-based classification (Wang et al., 2024b) have focused on modeling uncertainty in the target (output) space, these methods do not explicitly represent

uncertainty over model parameters. In contrast, our Epistemic Wrapper framework introduces a second-order uncertainty representation in the parameter space via belief functions. By wrapping posterior distributions derived from BNNs, our approach captures variability before the prediction stage, allowing for richer epistemic modeling. This can be particularly beneficial in scenarios with limited training data or when encountering out-of-distribution (OoD) inputs.

Approaches that operate solely in the output space such as decompositions of predictive uncertainty via proper scoring rules, Bregman divergences, calibration methods, or generalized bias-variance analyses (Kotelevskii et al., 2024; Ahdritz et al., 2024; Gruber & Buettner, 2022) provide important insights into the behaviour of predictive distributions. However, these techniques characterize EU only indirectly, after uncertainty has already propagated through the model. In contrast, our framework addresses uncertainty at its source by representing and enriching parameter-space posteriors before they influence predictions. This distinction allows us to capture model-level uncertainty in a principled way, offering a complementary perspective to output-based decompositions. By integrating belief functions and Dirichlet representations directly in parameter space, our method provides a prior-agnostic and interpretable mechanism for epistemic uncertainty quantification, while remaining fully compatible with existing Bayesian neural networks.

We stress that parameter-space and target-space approaches are not substitutes but complementary perspectives. Target-space methods characterize predictive uncertainty directly, while parameter-space methods quantify the model's internal epistemic state before it influences predictions. Our experiments show that Epistemic Wrapper consistently improves on standard downstream tasks such as OoD detection, calibration (ECE), and likelihood evaluation (NLL), providing strong empirical evidence that parameter-space epistemic modeling yields practical benefits in addition to its theoretical grounding. In summary, Epistemic Wrapper serves as a principled complement to target-based methods, enriching parameter-space uncertainty modeling. Importantly, our method is not directly comparable to target-based approaches such as EDL or ensembles, because they operate in a fundamentally different space.

# E    EPI-WRAPPER FOR REGRESSION TASKS

In principle, the Epistemic Wrapper is not restricted to classification tasks. The proposed framework models epistemic uncertainty in the parameter space of Bayesian Neural Networks via belief functions, which is independent of the nature of the output space. The only requirement for extending the method to regression tasks is the availability of a suitable Interval Neural Network (INN) or similar regression architecture capable of consuming interval-valued parameters during inference. While we have focused on classification tasks in this work, we believe that extending the approach to regression is a natural and promising direction for future work.

