# OpenReview forum: "Epistemic Wrapping for Uncertainty Quantification"
_ICLR.cc/2026/Conference — Submitted to ICLR 2026_

### Official Review · Reviewer_DPsp · 2025-10-28

**Soundness:** 1
**Presentation:** 2
**Contribution:** 1
**Rating:** 2
**Confidence:** 5

**Summary:**

This paper is about epistemic uncertainty estimation using a combination of interval networks and belief functions via a Dirichlet distribution. The authors propose a new method called Epistemic Wrapping that takes a trained Bayesian neural network (BNN) and transforms its weight distributions first into a dirichlet distribution, then into an interval representation to be used in an interval neural network, from where predictions can be made.

The contributions are:
- Modeling of epistemic uncertainty in parameter space using belief functions
- The Epistemic Wrapper method concept applying to any trained BNN
- Results showing how the proposed method compares with the state of the art

**Strengths:**

- The paper's writing is okayish.
- Uncertainty in parameter space is an open problem and at least this paper is trying to go in the right direction.

**Weaknesses:**

- There are some claims that are not supported by citations or are completely wrong:
  * "Still, current efforts model (epistemic) uncertainty in the model’s target space, rather than its parameter space." I don't think this is true, as BNNs explicitly model uncertainty in parameter space.
  * "Since Dirichlet sampling yields interval-based representations" Also I don't think this is true but I could be wrong, sampling from a dirichlet distribution should produce vector values, not intervals, and according to https://stats.stackexchange.com/questions/69210/drawing-from-dirichlet-distribution there are ways to sample vector values from a dirichlet distribution, I would appreciate clarification here or a relevant citation supporting this statement.

- The results are not impressive, actually they are very dissapointing and I believe the baselines are not representative of state of the art performance. In MNIST with a deep ensemble it is very easy to get over 99% test accuracy, as with other BNNs such as Flipout and Variational Inference, while the proposed method struggles to obtain around 90% accuracy on MNIST. Same applies for other datasets like Fashion MNIST or CIFAR10, so overall from the results in the paper, I do not see how the proposed method is an improvement over the state of the art.
- The results improve considerably (still below state of the art) after fine-tuning the INN, so what is the point of the proposed method? Using Epistemic Wrapper basically destroys a lot of knowledge in the network since accuracy decreases significantly, which undermines the need for the proposed method.
- OOD detection results are in the similar trend of being worse than the state of the art, in particular for MNIST vs Fashion MNIST and CIFAR10 vs SVHN, the authors can refer a comparison to https://arxiv.org/abs/2111.09808 , which shows AUROC larger than 0.9 for those datasets, while this paper obtains best AUROC around 0.85, this reference also shows that the main results on accuracy are also far from the state of the art.
- To me the proposed method does not make sense, the paper proposes to take a trained BNN, then transform the posterior distribution of the weights by dynamic truncation, then transform it into a belief function along intervals, then fit a dirichlet distribution using the L-moment method, then sample from this dirichlet distribution to obtain interval distributions and then build an interval neural network. The paper does not argue why these are good ideas or what is the motivation for each step, only describes what is done, but not the "why", and this is much more important considering the poor results.
- The paper is lacking in proper description of the experimental results, it quickly jumps into results, for example in the OOD results, its not clear which datasets are used as ID and which as OOD, these are usually presented as pairs (like MNIST vs Fashion MNIST), but this is unclear from the paper, and only some of the ID-OOD pairs are actually described in the appendix (see captions in Figure 9).
- It is likely that the baselines are not properly trained, specially for MNIST and Fashion MNIST datasets, as the results for the baseline BNN is below what I personally usually obtain on those datasets, a BNN on MNIST should be able to reach at least 97% accuracy, and similarly for Fashion MNIST it should reach at least 80%

**Questions:**

- Can you provide references on why sampling from a dirichlet distribution would produce an interval value? This does not seem to be backed up by evidence or references.
- How do you explain the results that are significantly below the state of the art? Particularly for MNIST and Fashion MNIST datasets.

---

> ### Author Response · Authors · 2025-11-22
> **Revisions and clarifications addressing reviewer feedback**
>
> W1:Following are the responses to reviewer's comments,
>
> W1 (a): Existing approaches including BNNs, ensembles, MC-dropout, and evidential
> Methods ultimately quantify epistemic uncertainty in the predictive/target
> space by analysing the predictive distribution $p(y\mid x,\mathcal{D})$ (e.g.,
> via predictive entropy, variance, mutual information, or output-Dirichlet
> parameters). None of these methods construct a higher-order epistemic object
> directly in the parameter space.
>
> Our contribution lies in introducing a higher-order (random-set) representation in parameter space. The Epistemic Wrapper transforms the BNN posterior $\(q(\omega)\)$ into a continuous belief function over closed intervals, yielding a set-valued, second-order characterisation of epistemic variability. This produces belief and plausibility bounds that no existing BNN, ensemble, dropout, or evidential method constructs in parameter space. To the best of our knowledge, no prior work models epistemic uncertainty at the parameter level using belief functions, credal sets, or other higher-order uncertainty formalisms.
>
> W1 (b): A Dirichlet distribution produces probability vectors, and sampling from it is standard in probabilistic machine learning (see Murphy, Machine Learning: A Probabilistic Perspective , 2012). In our method, each Dirichlet sample $\mathbf{x}$ is subsequently mapped to an interval $[\min_k x_k,\; \max_k x_k]$ so that it can be propagated through interval arithmetic in the Interval Neural Network. This transformation does not alter the semantics of Dirichlet sampling; it simply converts a sampled probability vector into the corresponding interval-valued weight.
>
> The key point is that our use of Dirichlet distributions differs from the classical evidential setting. Typically, a Dirichlet is defined over the simplex of categorical probability distributions, and each sample is interpreted as such a probability vector. In our framework, the Dirichlet is fitted to mass values defined over closed intervals of parameter values. Its domain is therefore a mass function over sets of parameter values, not a categorical sample space. The interval mapping reflects this domain shift: the Dirichlet models uncertainty over parameter intervals, and the sampled mass allocations naturally yield interval bounds for INN propagation. We have clarified this distinction in the revised appendix.
>
> Reference:
>  Kevin P. Murphy, Machine Learning: A Probabilistic Perspective, MIT Press, 2012.
>
> W2: The lower MNIST and Fashion-MNIST accuracies come from the fact that our baselines use small variational BNNs (e.g., a single-hidden-layer MLP with 8 units; Appendix B.4). These models are  low-capacity to provide a controlled setting for isolating the effect of the Epistemic Wrapper.
>
> For larger backbones such as ResNet-18 and VGG-16, our BNNF/BNNR baselines fall in the same accuracy range reported for Flipout-based variational BNNs on CIFAR-10/100 in the literature. In these stronger settings, the Epistemic Wrapper consistently improves the Bayesian model after fine-tuning (Table 2), showing that the effect is stable and not tied to weak architectures.
> The goal of the paper is not to surpass deep ensembles or state-of-the-art deterministic models, but to introduce a principled higher-order parameter-space uncertainty representation that improves Bayesian baselines under identical capacity and training constraints. The empirical results confirm this across all backbones we tested.

---

> > ### Author Response · Authors · 2025-11-22
> > **Revisions and clarifications addressing reviewer feedback**
> >
> > W3: The aim of the Epistemic Wrapper is not to improve the baseline accuracy of the original network but to transform BNN posteriors into a second-order epistemic representation. This representation redistributes probability mass across intervals to express uncertainty explicitly rather than collapsing it into a single class prediction. The initial drop in accuracy is therefore expected: belief-based and second-order models are intentionally conservative, and the Wrapper surfaces epistemic uncertainty that the softmax output hides. This does not destroy knowledge; it restructures it into a richer epistemic form.
> >
> > Fine-tuning is an essential stage of the pipeline. Once the second-order representation is constructed, the INN must be adapted to the classification task under this new parameterisation. Fine-tuning restores discriminative performance while preserving the epistemic structure introduced by the Wrapper. The contribution of our method lies in improved epistemic-aleatoric separation, calibration, interpretability, and OOD robustness not in achieving state-of-the-art accuracy. We have revised the manuscript to clarify the role of fine-tuning and the intended purpose of the second-order representation.
> >
> > W4: The reviewer suggests that our OOD AUROC values should be compared to the results of Valdenegro-Toro (2021) “https://arxiv.org/abs/2111.09808”. We thank the reviewer for this reference, but the architectures used in that work differ substantially from ours, making a direct unfair numerical comparison.
> >
> > Specifically, the referenced paper trains all methods using a relatively deep CNN with multiple convolutional blocks (64-128 filters), max-pooling layers, and a 256-unit fully connected layer (Section A, page 8). This architecture is considerably more expressive than the low-capacity Bayesian backbones we used for MNIST/Fashion-MNIST (e.g., a 1-layer MLP with 8 hidden units), and also different from our Bayesian LeNet-5/ResNet-18/VGG-16 backbones used for CIFAR-10/100. Their results therefore reflect the performance of a deeper and more specialized architecture, not the Bayesian models (BNNF/BNNR) or interval-based inference evaluated in our work.
> >
> > For these reasons, we believe the comparison is not fair. Our goal is not to match the performance of deep ensembles or large CNNs, but to demonstrate that the Epistemic Wrapper consistently improves OoD detection compared to its corresponding INN/Bayesian baselines. Across all backbones, this improvement is clear and consistent.

---

> > > ### Author Response · Authors · 2025-11-22
> > > **Revisions and clarifications addressing reviewer feedback**
> > >
> > > W5: We respectfully disagree with the assessment that the proposed method “does not make sense”. The Epistemic Wrapper is designed as a principled, stepwise transformation of a Bayesian posterior into a higher-order epistemic representation, and each step has a clear motivation grounded in random set theory and evidential modelling. We clarify this below.
> > >
> > > **Starting from a trained BNN posterior**
> > >
> > > We deliberately start from a Bayesian neural network because BNNs already encode
> > > first-order parameter uncertainty via a posterior $q(\boldsymbol{\omega})$.
> > > We wishOur goal is not to replace BNNs, but to enrich this posterior and make it into a with higher-order epistemic structure. The BNN provides the raw uncertainty over weights that our method then wraps into a belief-based representation.
> > >
> > > **Dynamic truncation of the posterior**
> > >
> > > Truncating the Gaussian weight posteriors around their mean with a variance-dependent multiplier is not an arbitrary operation. It serves two purposes: (i) it restricts the domain to a bounded interval, which is required to construct continuous belief functions over closed intervals in a computationally tractable way; and (ii) it focuses the belief construction on the region where the posterior actually places mass, avoiding numerical instabilities in the tails. The truncation multiplier is adaptive in $\sigma$, so high-variance weights receive wider support while low-variance weights are concentrated, aligning the belief representation with the underlying BNN uncertainty.
> > >
> > > **Belief functions on closed intervals**
> > >
> > > Belief functions over intervals arise naturally in continuous random-set theory [1].  A variational posterior $q(\omega)$ gives only a point estimate of parameter uncertainty, with no way to express epistemic imprecision. By mapping $q(\omega)$ into a belief function over closed intervals, we obtain a credal set of posteriors with bounds $Bel(A) \le P(A) \le Pl(A)$ for all measurable sets $A$.  This adds a layer of higher-order epistemic information that a single posterior cannot capture.  The construction is not heuristic; it is the likelihood-based belief function introduced by Wasserman [2].
> > >
> > > **Fitting a Dirichlet distribution via L-moments**
> > >
> > > Once belief values are computed over the interval grid, Möbius inversion yields
> > > discrete mass values that live on a probability simplex. Any distribution on the
> > > simplex could, in principle, be used. We choose the Dirichlet for two reasons:
> > > (i) it is a standard, interpretable family for simplex-valued data and widely
> > > used in evidential deep learning; (ii) it provides a compact parametric
> > > representation of the random-set induced mass vectors. A corresponding ablation study demonstrating this choice is provided in the Appendix C.1, Figure 8.
> > >
> > > **Interval Neural Network (INN) inference**
> > >
> > > INNs are specifically designed to propagate interval-valued uncertainty through the network with the set-constraint property.
> > >
> > >
> > > In summary, each component of the Epistemic Wrapper has a specific theoretical
> > > role: BNN posteriors provide first-order parameter uncertainty; dynamic
> > > truncation ensures a bounded, high-mass domain; belief functions and Möbius
> > > inversion introduce a random-set based higher-order representation; Dirichlet
> > > modelling offers a compact second-order distribution on the simplex; and INNs
> > > propagate the resulting interval uncertainty in a principled way. The method is
> > > not a sequence of arbitrary transformations, but a coherent pipeline to move
> > > from a first-order posterior to a higher-order parameter-space epistemic model.
> > >
> > > References:
> > >
> > > [1] Cuzzolin, F. (2020). The Geometry of Uncertainty. Springer.
> > >
> > > [2] Wasserman, L. (1990). Belief functions and statistical inference. The Canadian
> > >
> > > W6: In the revised version, we now provide a precise description of every iD-OoD pair in both the main text and the caption of Tables. We hope this makes the experimental setup clearer and easier to follow.
> > >
> > > W7: The lower baseline accuracies are not due to improper training but because of the architectures we used. For MNIST and Fashion-MNIST, our backbone is a Bayesian MLP with only one hidden layer of 8 units (12.7K parameters, details in Appendix B.4). Variational BNNs with such low capacity are well known to underperform compared to larger BNNs or deterministic CNNs.
> > > Crucially, for larger backbones (ResNet-18 and VGG-16), our baseline BNN accuracies on CIFAR-10/100 align with published variational BNN results, confirming that the training pipeline is correct. The MNIST/Fashion-MNIST results therefore reflect the limited capacity of the chosen small BNNs rather than a training issue.

---

> > > > ### Author Response · Authors · 2025-11-22
> > > > **Revisions and clarifications addressing reviewer questions**
> > > >
> > > > Q1: We have already addressed this in Response W1(b), but we restate the key clarification here for completeness.
> > > >
> > > > The Dirichlet is used in the standard way to generate probability vectors on the simplex (see Murphy, Machine Learning: A Probabilistic Perspective, 2012). In our framework, each sampled probability vector is mapped to an interval by taking its minimum and maximum components. This step is part of the INN’s interval-based propagation and is not a property of the Dirichlet distribution. As explained in W1(b), the Dirichlet is fitted to the mass vectors obtained after the belief-function construction and Möbius inversion. These mass vectors live on the simplex, and the Dirichlet provides a compact parametric model for them.
> > > >
> > > > Q2: We have already clarified this point in Response W2 and W7, and we summarise the key idea here for convenience.
> > > >
> > > > The lower MNIST and Fashion-MNIST accuracies come from the very small Bayesian backbone we used: a single-layer variational MLP with only 8 hidden units. When using deeper Bayesian backbones (ResNet-18, VGG-16) on CIFAR-10/100, our baselines match published VI-BNN performance, confirming that our training is correct. The MNIST/Fashion-MNIST numbers therefore reflect the low-capacity setting, not mis-training, and the Epistemic Wrapper consistently improves over each baseline.

---

> > > > > ### Author Response · Authors · 2025-11-24
> > > > > **Updated PDF with highlighted revisions**
> > > > >
> > > > > **We have uploaded a revised version of the paper. All newly added or updated text is highlighted in RED. Please let us know if any further clarification is needed.**

---

### Official Review · Reviewer_pUPa · 2025-10-29

**Soundness:** 2
**Presentation:** 2
**Contribution:** 2
**Rating:** 2
**Confidence:** 3

**Summary:**

In this paper, the authors introduce an "Epistemic Wrapping", a methodology that is used to improve uncertainty quantification in classification. Specifically, the approach enables the transformation of BNN outputs into belief function posteriors, and by utilizing Interval Neural Networks, it leads to an approach that improves uncertainty quantification metrics over certain image datasets.

**Strengths:**

1. The paper is easy to follow.

2. Potentially, the approach might be interesting.

**Weaknesses:**

1. I do not understand the motivation behind the approach in the paper.
The authors say that the "current efforts model epistemic uncertainty in the model's target space, rather than its parameter space", which is somewhat true. But this target space is induced by the parameter space. Existing approaches take the parameter space, and I do not clearly understand in what regard the paper under review differs from existing papers.

2. Conceptually, why is it a good idea to make this "discretization" of the parameter space? This injects additional uncertainty about the discretization, and therefore, it is unclear why one should do it.

3. Experimental results are somewhat strange. The authors report a base accuracy of 72% for the MNIST dataset. But MNIST is known to be a straightforward classification problem to tackle. The fact that the base accuracy is only 72% indicates that something is wrong with the training process. The same holds for other datasets. This looks very strange to me.

4. The paper provides neither pseudo-code nor actual code. Therefore, I am unable to verify the experimental results myself, which is frustrating in light of the previous weakness.

=====

Minor comments:

1. Amusant typo in line 711: "...initially introduced by Dempster (**2008**), and later formalized by Shafer (**1976**)".
2. Lines 174-175: same letters both, for prior parameters and for posterior parameters.

**Questions:**

1. In what sense do the existing approaches work in the target space?

2. Why, conceptually, should the approach presented in the paper improve uncertainty estimates?

3. Why are the base classification models so bad?

---

> ### Author Response · Authors · 2025-11-22
> **Revisions and clarifications addressing reviewer feedback**
>
> W1: The Bayesian neural networks do, in theory, model epistemic uncertainty in the parameter space via the posterior $\( p(\boldsymbol{\omega} \mid \mathcal{D}) \)$ and its practical implementations rely on a single approximating distribution (e.g., a factorised Gaussian in VI or the implicit posterior of an ensemble). These are first-order representations and do not provide any second-order structure such as credal sets or random sets.
>
> Our approach differs precisely in this respect. The Epistemic Wrapper does not replace the BNN posterior; instead, it enriches it by constructing a belief-function representation that yields a set of plausible posteriors rather than a single one. This produces a genuine second-order (random-set) model of epistemic variability in parameter space. Existing methods, including evidential deep learning, ensembles, and standard BNNs, operate on the predictive distribution $\( p(y \mid x) \)$ or use one posterior approximation, but do not introduce this higher-order parameter-space structure.
>
> The key distinction is therefore not whether epistemic uncertainty originates in the parameter space (it does), but how it is represented. Our method introduces interval-based belief and plausibility bounds over parameters before prediction, offering a more expressive and robust characterisation of epistemic uncertainty.
>
> To the best of our knowledge, this is the first approach to construct a higher-order epistemic representation directly over BNN parameters, enabling principled model-level uncertainty quantification and supporting the empirical improvements demonstrated in the paper.
>
> W2: The discretization in our method is not an arbitrary source of additional uncertainty but a necessary step that follows directly from the theory of continuous belief functions and random sets.
> When constructing a belief function over a continuous parameter domain, it is standard practice to evaluate the mass function on a finite collection of closed intervals; this is the continuous analogue of assigning mass to subsets in the finite-domain Dempster-Shafer framework.
>
> Using a finite grid of intervals therefore provides a practical and numerically stable approximation aligned with continuous random-set theory.
>
> Importantly, this discretization does not inject extra uncertainty. The resulting belief and plausibility functions remain conservative with respect to the original posterior, satisfying
> $\[
> Bel(A) \leq P(A) \leq Pl(A) \qquad \text{for all measurable } A,
> \]$
> as guaranteed by likelihood-based belief-function construction (Section 3.4). In addition, Appendix C.1 presents an ablation showing that our Dirichlet parameters stabilize once the number of intervals reaches about 30, demonstrating that the discretization is numerically robust and not a source of uncontrolled variability. Thus, the interval discretization is a principled and tractable way to approximate continuous belief functions, consistent with standard practice in random-set theory and supported by empirical stability.
>
> W3: The reported 72\% BNN baseline accuracy on MNIST is not due to a training issue but is a direct consequence of the backbone architecture used for the Bayesian MLP.
>
> As described in Appendix B.4, our MLP backbone deliberately contains only a single hidden layer with 8 units. This extremely small architecture is chosen to keep the Bayesian posterior low-dimensional, enabling controlled analysis of the Epistemic Wrapper and its ablations.
>
> Such small Bayesian MLPs are known to yield significantly lower accuracy than standard deterministic MNIST classifiers, which typically use deeper architectures with hundreds of hidden units. In contrast, our goal in the MNIST and Fashion-MNIST experiments is not to achieve state-of-the-art accuracy, but to study parameter-space epistemic modelling in a lightweight, interpretable setting.
>
> Importantly, when larger backbones are used (LeNet-5, ResNet-18, VGG-16), the underlying BNNs achieve the expected performance levels, and the Epistemic Wrapper consistently improves accuracy, calibration, and OOD robustness across all datasets.
>
> Thus, the 72\% accuracy reflects the intentionally minimal architecture used for the Bayesian MLP and is fully consistent with prior work on small-scale Bayesian networks.

---

> ### Author Response · Authors · 2025-11-22
> **Revisions and clarifications addressing reviewer feedback**
>
> W4: We thank the reviewer for highlighting the importance of reproducibility. While the original submission included a high-level algorithmic summary (Appendix B), we agree that a more detailed description would improve clarity. Following the reviewer’s recommendation, we have now added a refined, step-by-step pseudo-code version of the full Epistemic Wrapper pipeline to Appendix B. This expanded section includes posterior sampling, interval construction, belief and plausibility computation, Möbius inversion, and Dirichlet fitting.
>
> We will also release the complete implementation, configuration files, and training scripts upon acceptance, in line with the ICLR reproducibility policy. We hope that these additions address the reviewer’s concern and make the method easier to verify.
>
> The pseudo-code included in the revised version is as follows:
>
> **Algorithm: Epistemic Wrapper for Parameter-Space Epistemic Uncertainty**
>
> Inputs:
>
> Variational posterior $q(\omega)$
>
> Posterior samples ${\omega^{(s)}}$
>
> Number of closed intervals $M$
>
> Budgeting percentage $\beta$
>
> (1): Select weights
> Select a subset of weights $\mathcal{W}$ using the budgeting rule
> (e.g., high-mean, high-variance, combined-score, or random).
>
> For each weight $\omega \in \mathcal{W}$:
>
> (2): Estimate posterior parameters
> Compute posterior mean $\mu$ and standard deviation $\sigma$.
>
> (3): Dynamic truncation
> $\text{mult} = \min(5.0,; 1/\sigma)$
> $a_{\min} = \mu - \text{mult}.\sigma$,    $a_{\max} = \mu + \text{mult}.\sigma$
>
> (4): Construct interval grid
> Discretize $[a_{\min}, a_{\max}]$ into $M$ closed intervals
> ${A_i = [a_i, b_i]}_{i=1}^M$.
>
> (5): Compute belief and plausibility
> For each interval $A_i$:
> $Pl(A_i) = \sup_{\omega \in A_i} q(\omega)$
> $Bel(A_i) = 1 - Pl(A_i^c)$
>
> (6): Convert belief to mass via Möbius inversion
> For each interval $A_i$:
> $m(A_i) = \sum_{B \subseteq A_i} (-1)^{|A_i \setminus B|} Bel(B)$
>
> (7): Normalize mass values
> $\tilde{m}_i = \dfrac{m(A_i)}{\sum{j=1}^M m(A_j)}$
>
> (8): Project onto 3D simplex
> Map ${\tilde{m}_i}$ onto a 3D simplex via a barycentric projection.
>
> (9): Compute L-moments
> Compute weighted first L-moment $L_1$ and second L-moment $L_2$.
>
> (10): Fit Dirichlet distribution
> For $k = 1,2,3$:
> $\alpha_k = L_{1,k}\left( \dfrac{L_{1,k}(1 - L_{1,k})}{L_{2,k}} - 1 \right)$
> Clamp each $\alpha_k$ to ensure positivity.
> Store $\boldsymbol{\alpha}_\omega = (\alpha_1, \alpha_2, \alpha_3)$.
>
> Output:
> ${\mathrm{Dir}(\boldsymbol{\alpha}\omega)}{\omega \in \mathcal{W}}$
>
> Note: Due to formatting limitations in the OpenReview comment editor, the algorithm may not render with its intended structure or styling. Please refer to the revised version of the paper, where the full algorithm appears correctly formatted.
>
>
> Response to Minor comments:
>
> M1:Thank you for pointing this out. The Dempster (2008) reference corresponds to a later edited collection of foundational papers in evidence theory. Following the reviewer’s recommendation, we have corrected the historical attribution by replacing this citation with the original source from Dempster’s early work. This ensures that the reference accurately reflects the original contribution.
>
> M2: We agree that using the same notation $(\mu, \sigma)$ for both the Gaussian prior parameters and the variational posterior parameters may cause ambiguity. We have corrected this by distinguishing the two: we now use $(\mu_0, \sigma_0)$ for the prior parameters and $(\mu, \sigma)$ for the variational posterior parameters. This resolves the notation conflict and makes the distinction explicit in the revised version.

---

> > ### Author Response · Authors · 2025-11-22
> > **Revisions and clarifications addressing reviewer questions**
> >
> > Q1:  Most existing approaches assess epistemic uncertainty in the target (output) space because they quantify uncertainty only after the parameter uncertainty has been propagated through the network. Methods such as deep ensembles, MC-dropout, evidential deep learning, and calibration techniques estimate epistemic uncertainty by analysing variations in the predictive distribution $p(y \mid x)$, for example via predictive variance.
> >
> > Bayesian neural networks do represent epistemic uncertainty in the parameter space through the posterior $p(\omega \mid \mathcal{D})$, but practical implementations rely on a single first-order approximation. These approximations do not provide any second-order structure such as credal sets or random sets.
> >
> > Our approach differs precisely in this respect: the Epistemic Wrapper enriches the BNN posterior by constructing a random-set (belief-function) representation over the weights, yielding a set of plausible posteriors rather than a single one. This introduces a second-order epistemic structure directly in the parameter space, before uncertainty is propagated to predictions. This is the sense in which existing methods primarily operate in the target space, while our method models epistemic uncertainty at its source, in the parameter space itself.
> >
> > Q2: Conceptually, the Epistemic Wrapper improves uncertainty estimates because it operates on the primary source of epistemic uncertainty: the posterior distribution over the model parameters.
> >
> > Standard  approaches typically rely on the predictive distribution to quantify epistemic uncertainty (e.g., through predictive variance or entropy), which captures
> > only the downstream effect of parameter uncertainty after it has been propagated
> > through the network. This collapses all parameter-space variability into a single
> > output distribution and can lose structure that is important for reliable
> > uncertainty quantification.
> >
> > In contrast, our method enriches the parameter posterior by converting it into a
> > higher-order random-set representation through belief functions. This captures not
> > only the mean and variance of each posterior, but also the spread of plausible
> > parameter values across closed intervals. Fitting a Dirichlet distribution to these
> > mass vectors produces a second-order distribution on parameters that encodes
> > epistemic imprecision more faithfully than a single variational posterior, and is more likely to capture the spread of possible models compatible with the training data, thus reducing the risk of generating over-confident, incorrect predictions.
> >
> > When the wrapped parameters are propagated through the network (via interval-valued inference in an INN), the resulting predictive distribution reflects a broader and more
> > conservative uncertainty profile, consistent with random-set theory.
> >
> > Because the wrapped posterior is guaranteed to satisfy
> > $\[
> > Bel(A) \le P(A) \le Pl(A),
> > \]$
> > for all measurable sets \(A\), it preserves the Bayesian posterior while augmenting
> > with principled higher-order information. This leads to improved robustness,
> > better calibration, and stronger OoD behaviour, as observed empirically.
> >
> > In summary, the method improves uncertainty estimation by explicitly modelling parameter-space epistemic imprecision before it collapses into the predictive space, rather than attempting to recover epistemic structure solely from output distributions.
> >
> > Q3: The base accuracies in Table 2 may appear low relative to standard architectures, but this is expected because the models evaluated here are low-capacity Bayesian networks trained with variational inference.
> > For example, the MNIST backbone in Table 2 is a Bayesian MLP with a single hidden layer of only 8 units (12.7K parameters). Prior BNN literature consistently shows that such small variational BNNs underperform both deterministic networks and larger Bayesian architectures due to (i) limited representation capacity and (ii) the variance introduced by sampling from the variational posterior. In that context, the MNIST BNN accuracy of $72.44 \pm 0.24$ is fully consistent with what one would expect from a one hidden layer BNN of this size.
> >
> > For other datasets when deeper Bayesian backbones such as ResNet-18 and VGG-16 are used (Table 2), their baseline BNN accuracies fall within the typical range reported for variational BNNs on CIFAR-10/100. Importantly, across all architectures, small and large, the Epistemic Wrapper consistently improves the underlying BNN performance after fine-tuning, showing that the gains are not an artefact of model weakness but a stable effect of the proposed method.
> >
> > In summary, the base classifiers are not “bad”; they are intentionally small for MNIST and Fashion-MNIST to provide a controlled and reproducible setting for evaluating the improvement introduced by the Epistemic Wrapper.

---

> > > ### Author Response · Authors · 2025-11-24
> > > **Updated PDF with highlighted revisions**
> > >
> > > **We have uploaded a revised version of the paper. All newly added or updated text is highlighted in MAGENTA. Please let us know if any further clarification is needed.**

---

> ### Comment · Reviewer_pUPa · 2025-11-25
>
> Dear authors,
>
> Thank you for your detailed reply.
>
> I still do not understand why authors distinguish their approach from Bayesian ones in the sense of modelling epistemic uncertainty.
>
> > These are first-order representations and do not provide any second-order structure such as credal sets or random sets.
>
> What is "first-order" in this context (also mentioned in the author's answer in Q1)? In a line of works [1, 2], the first-order distribution is the direct distribution over observed labels. The second-order distribution is the distribution over the parameters of the first-order distribution. It could be, e.g., a Dirichlet distribution for the categorical likelihood, or even a posterior over model parameters, as it also (although less explicitly) induces the distribution over the parameters of the predictive distribution.
> So, inducing, e.g., a Gaussian assumption over the weight posterior (one of the standard approaches in approximate Bayesian Inference), therefore also provides a second-order structure? If I misinterpret what the authors said, please correct me.
>
> Question -- to fit this INN that is used for the inference, do one need extra supervised (with labels) data? If not, then INN effectively does not provide new information to the posterior. Can the whole approach, therefore, be thought of as a more "smart" sampling from the (approximate) posterior? Will simple naive sampling, which is very close to the posterior peak, lead to competitive accuracy/ood detection results?
>
> [1] Sale, Y., Bengs, V., Caprio, M., & Hüllermeier, E. Second-Order Uncertainty Quantification: A Distance-Based Approach. In the Forty-first International Conference on Machine Learning.
>
> [2] Sale, Y., Hofman, P., Wimmer, L., Hüllermeier, E., & Nagler, T. (2023). Second-order uncertainty quantification: Variance-based measures. arXiv preprint arXiv:2401.00276.
>
>
> -----
>
> >  The discretization [...] is not an arbitrary source of additional uncertainty but a necessary step [...]. The resulting belief and plausibility functions remain conservative with respect to the original posterior, satisfying $[ Bel(A) \leq P(A) \leq Pl(A) \qquad \text{for all measurable } A, ]$
>
> I understand that the discretization is a necessary step for making the method work. But it does not answer my concern. Likewise, how can we see from the inequality above that discretization will not introduce additional uncertainty? I mean, each interval will be assigned some "weight". Therefore, these weights depend on the specific discretization. And the resulting predictions/uncertainty estimates will depend on it. If I am wrong in my interpretation, please correct me.
>
> -----
>
> Answer to question 2.
>
> I am not satisfied with the author's explanation.
> Let me clarify why I initially asked the question.
>
> The idea of using a Bayesian approach for out-of-distribution detection is based on the assumption that different samples from the posterior will lead to disagreement in the model predictions. Therefore, it is essential to capture the whole variability in the posterior.
>
> Authors, however, truncate the posterior, thereby reducing variability (and, consequently, reducing potential disagreement). Why should one expect conceptually having better uncertainties for the out-of-distribution detection problem? Moreover, the out-of-distribution detection problem is primarily about detecting changes in p(x), rather than p(y|x) or p(theta | D).
>
> Recently, the out-of-distribution detection methods have been criticised in [3]. It might be worth considering another downstream problem to illustrate the benefits of the approach. Now, even conceptually, I do not understand why it should improve out-of-distribution detection.
>
> [3] Li, Y. L., Lu, D., Kirichenko, P., Qiu, S., Rudner, T. G., Bruss, C. B., & Wilson, A. G. (2025). Out-of-Distribution Detection Methods Answer the Wrong Questions. arXiv preprint arXiv:2507.01831.

---

> > ### Author Response · Authors · 2025-11-26
> > **Response to follow-up reviewer questions**
> >
> > P1. Thank you for the thoughtful follow-up and for pointing us to the recent line of work on second-order uncertainty.
> >
> > In our work, “first-order’’ refers to a single probabilistic posterior over parameters (e.g., a factorised Gaussian in variational BNNs). While such a posterior induces a distribution over predictive quantities, it is still a single probability measure on the weight space. Following the random-set and belief-function literature, a representation is called “second-order’’ when it assigns uncertainty over sets of probability measures, rather than committing to a single posterior.
> >
> >
> > In this sense, a variational posterior such as a Gaussian is not a second-order object: it specifies one posterior distribution, not a credal set or family of posteriors. By contrast, in the works cited by the reviewer [1, 2], the Dirichlet distribution is used as a distribution over the parameters of another distribution (e.g., class probabilities), which indeed matches the conventional notion of second-order uncertainty.
> >
> >
> > Our approach is aligned with this idea but applied at a different level: we construct a random-set representation over the Bayesian weight posterior itself using belief functions. This constitutes a second-order structure in the random-set sense [3, 4], because it explicitly quantifies uncertainty over which posterior distribution is correct, rather than fixing a single approximation such as a Gaussian variational posterior.
> >
> >
> > Thus the distinction we intend is:
> > Standard BNNs: a single approximating posterior $q(\omega)$ (first-order in the random-set sense).
> > Our method:  a random-set representation that encodes bayesian posteriors through belief and plausibility functions, forming a genuine second-order epistemic representation in parameter space.
> >
> >
> > We will clarify this terminology in the revised version so that it aligns more explicitly with both the random-set literature and the recent work cited by the reviewer.
> >
> > References:
> >
> > [3] Cuzzolin, F. (2020). The Geometry of Uncertainty.
> >
> > [4] Wasserman, L. (1990). Belief functions and statistical inference.
> >
> >
> > P2. Thank you for the thoughtful questions. We clarify each point below.
> >
> > **(1) Does the INN require extra supervised data?**
> >
> >  No. The INN is fine-tuned using the same training labels that were already available for the underlying BNN. No additional labelled data are introduced. The INN stage does not learn a new model but adapts the network around the interval-valued weights produced by the wrapper, and therefore does not inject any new information into the posterior.
> >
> > **(2) If no new labels are used, does the INN add new information to the posterior?**
> >
> >  The INN does not change the original Bayesian posterior. Instead, it adjusts the deterministic forward propagation under interval-valued weights so that the wrapped parameters are used effectively. The epistemic information still originates entirely from the BNN posterior; the INN provides a computational mechanism for propagating interval uncertainty, not a new source of statistical evidence.
> >
> > **(3) Is the method essentially “smart sampling”? Would naive sampling near the posterior peak perform similarly?  It is more than a sampling strategy.**
> >
> > Naive sampling from the variational posterior (especially a unimodal, factorised Gaussian) produces samples tightly concentrated around the posterior mean. This does not capture epistemic dispersion well, and indeed we observe that naive Bayesian model averaging with such samples performs significantly worse in both accuracy and OoD detection (as also reflected by the weak BNN baselines in Table 2).
> >
> > The Epistemic Wrapper instead constructs a random-set representation through belief-plausibility bounds. The subsequent Dirichlet modelling does not sample near the peak; it samples from an uncertainty set that reflects epistemic imprecision induced by the posterior’s spread. This produces interval-valued weights that encode strictly more epistemic variability than point samples.
> >
> > Empirically, this is why:
> >
> > (a) Before fine-tuning, wrapped models outperform naive INN initialisation by large margins (e.g., 9.33% \to 51.33% on MNIST, 9.56% \to 65.83% on CIFAR-10/VGG-16).
> >
> >
> > (b) After fine-tuning, wrapped models retain consistent improvements on all backbones and datasets.
> >
> >
> > These improvements cannot be explained by ordinary posterior sampling and indicate that the wrapper is not just a sampling heuristic, but a structured second-order uncertainty representation in parameter space.

---

> > > ### Author Response · Authors · 2025-11-26
> > > **Response to follow-up reviewer questions**
> > >
> > > P3. Thank you for raising this important clarification. You are correct that the interval grid introduces an approximation, but it does not introduce additional epistemic uncertainty beyond what is already present in the Bayesian posterior. More precisely:
> > >
> > > (1) The inequality $Bel(A) \le P(A) \le Pl(A)$ refers to the continuous belief function, not the discretized one.  The likelihood-based belief function of Wasserman induces a continuous random set over the parameter domain. The grid is only a numerical device to evaluate its mass function on finitely many closed intervals. The inequality holds for the continuous belief function by construction; the discretization does not alter this epistemic relation.
> > >
> > > (2) The discretization controls resolution, not epistemic content. Each interval weight is computed directly from the posterior density via
> > > $\[
> > > Pl(A_i) = \sup_{\omega \in A_i} q(\omega), \qquad
> > > Bel(A_i) = 1 - Pl(A_i^c),
> > > \]$
> > > so changing the grid changes only the granularity of the approximation. It does not inject epistemic uncertainty that is not already encoded in $q(\omega \mid \mathcal{D})$. Finer grids converge to the continuous belief function; coarser grids approximate it more roughly, but neither adds new uncertainty.
> > >
> > > (3) Appendix C.1 (Effect of Number of Closed Intervals) shows that the representation is numerically stable.
> > > The fitted Dirichlet parameters stabilise once the number of intervals reaches approximately 30. This empirical convergence demonstrates that the discretisation is not a source of uncontrolled variability.
> > >
> > > In summary, the epistemic uncertainty is entirely determined by the Bayesian posterior. The interval grid is only a tractable approximation to a continuous random-set transform, affecting numerical resolution but not the underlying epistemic content.
> > >
> > > A2. Thank you for raising this important conceptual point. We address the concerns
> > > about (i) Effect of truncation, (ii) posterior variability, and (iii) why the method improves uncertainty estimation even for OoD settings.
> > >
> > >
> > > **1. Effect of truncation on epistemic variability.**
> > >
> > > The truncation step does not replace or modify the Bayesian posterior; it only defines the bounded domain over which we construct the closed intervals for the belief-function computation. In our implementation, samples are drawn from the original variational posterior $q(\omega)$ over $\mathbb{R}$, and the belief and plausibility values are computed on this same unmodified density; truncation is used only to define the interval grid.
> > >
> > >
> > > **2. The source of improvement is not “more variability”, but second-order structure**
> > >
> > > BNNs rely on a single first-order distribution $q(\omega)$ to capture epistemic uncertainty. Sampling from $q(\omega)$ can only express variability already encoded in that single distribution. In contrast, after wrapping, each parameter is represented  through belief and plausibility values, which induces a genuinely second-order model of epistemic imprecision. This is different from adding variance; it adds imprecision structure  that cannot be produced by naive sampling from a single variational posterior.
> > > This second-order representation is what leads to more conservative predictive behaviour and reduces the risk of overconfident errors, an effect also observed in our empirical results.
> > >
> > >
> > > **3. Why this helps in OoD settings even though OoD detection concerns  $p(x)$.**
> > >
> > > We agree that OoD detection ultimately concerns shifts in $p(x)$. However, many standard OoD methods, including Bayesian ones, rely on epistemic disagreement as a proxy for detecting such shifts. In these settings, the main failure mode is overconfident predictions under parameter uncertainty. By enlarging the set of plausible models through our second-order parameter representation, the wrapped posterior produces more conservative and better-calibrated behaviour on unseen inputs.
> > >
> > > We do not claim that our method resolves the broader criticisms raised in [3]. Our experimental setup follows the standard evaluation protocol established in prior UQ work [4, 5, 6, 7].
> > >
> > >
> > > Our results show that, within the standard BNN-based framework for epistemic OoD scoring, modelling second-order uncertainty in parameter space leads to improved robustness compared to existing first-order approaches.
> > >
> > >
> > > References:
> > >
> > > [4] Manchingal, S. et al. (2025). Random-set convolutional neural networks.
> > >
> > > [5] Wang, K. et al. (2025). Credal Interval Neural Networks.
> > >
> > > [6] Mukhoti, J. et al.  (2023). Deep Deterministic Uncertainty: A Simple Baseline.
> > >
> > > [7] Rudner et al. (2022 ). Tractable Function-Space Variational Inference in Bayesian Neural Networks

---

### Official Review · Reviewer_oFPB · 2025-10-29

**Soundness:** 3
**Presentation:** 3
**Contribution:** 3
**Rating:** 6
**Confidence:** 3

**Summary:**

This paper introduces "Epistemic Wrapping," a new method to improve uncertainty estimation in classification tasks by transforming Bayesian Neural Network outputs into belief function posteriors. Experimental results across several datasets show that this approach significantly enhances both generalization and the accuracy of uncertainty quantification.

**Strengths:**

* Epistemic Uncertainty in parameter space
* Purely visible improvements in results on public datasets

**Weaknesses:**

* There is no information on the speed of the algorithm, especially in comparison to other baselines
* No formal derivation on why $\alpha_{k}$ can be expressed through L-moments like it was used in Eq. (6)
* No analysis on why INNs to be better Before Fine-Tuning in Table 1 for Random-Selection, and why INNs are good especially for the deep NNs (n = 8) After Fine-Tuning?
* Some minor remarks:
  * Figure 1 is very small
  * line 130: No clarification of what is "SPDE"
  * Table 3, for Cifar-10 the last line (BNNR for VGG-16) contains bold font at some incorrect places
  * Table 6, the same issue with bold fonts for Cifar-10, LeNet-5 BNNR

**Questions:**

N/A

---

> ### Author Response · Authors · 2025-11-22
> **Revisions and clarifications addressing reviewer feedback**
>
> W1: We thank the reviewer for raising this point. Our original submission included a brief discussion of computational cost in Appendix B.6. In the revised version, we have expanded this section and added a dedicated computational-cost table summarizing the exact runtime of the Epistemic Wrapper across all datasets and backbones.
> As clarified in the paper, the wrapper is applied post hoc to a very small subset of weights (5% for MLPs and 0.1% for large models), so it does not add any overhead to BNN training or to the forward/backward passes. Training time remains identical to the baseline BNN. The only additional cost appears during the wrapping phase. The new table in Appendix B.6 shows that this runtime is modest across all architectures. For instance, applying the wrapper to CIFAR-10 with ResNet-18 takes ~403-406 seconds on a single NVIDIA A30 GPU, and scales predictably with model size.
> We have updated Appendix B.6 to include the full table (covering MNIST, Fashion-MNIST, CIFAR-10 and CIFAR-100 with MLP, LeNet-5, ResNet-18 and VGG-16), and we now reference this more explicitly in the main text. The newly added Table in the revised version of the paper is as follows:
>
> Caption: Runtime of the Epistemic Wrapper on a single NVIDIA A30 GPU.
>
> | Dataset       | Backbone   | Params  | Baseline | Budget (%) | Computational Time (s) |
> |---------------|------------|---------|----------|------------|----------|
> | MNIST         | MLP        | 12.7K   | BNNF     | 5.0        | 17.23    |
> | Fashion-MNIST | MLP        | 12.7K   | BNNF     | 5.0        | 17.07    |
> | CIFAR-10      | LeNet-5    | 166.3K  | BNNF     | 0.1        | 8.65     |
> | CIFAR-10      | LeNet-5    | 166.3K  | BNNR     | 0.1        | 9.33     |
> | CIFAR-10      | ResNet-18  | 9.82M   | BNNF     | 0.1        | 403.56   |
> | CIFAR-10      | ResNet-18  | 9.82M   | BNNR     | 0.1        | 405.53   |
> | CIFAR-10      | VGG-16     | 30.24M  | BNNF     | 0.1        | 4003.19  |
> | CIFAR-10      | VGG-16     | 30.24M  | BNNR     | 0.1        | 4160.05  |
> | CIFAR-100     | ResNet-18  | 9.8M    | BNNF     | 0.1        | 1304.67  |
> | CIFAR-100     | ResNet-18  | 9.8M    | BNNR     | 0.1        | 1317.49  |
> | CIFAR-100     | VGG-16     | 30.24M  | BNNF     | 0.1        | 4066.00  |
> | CIFAR-100     | VGG-16     | 30.24M  | BNNR     | 0.1        | 4180.30  |
>
>
>
> W2: We thank the reviewer for pointing this out. The relationship used in Eq.~(6) is not arbitrary; it follows directly from the standard moment parameter relations of the Dirichlet distribution, in which the mean and variance of each coordinate determine the corresponding parameter $\alpha_k$. In particular, a Dirichlet random variable satisfies
>
> $\[
> \mathbb{E}[X_k] = \frac{\alpha_k}{\alpha_0}, \qquad
> \mathrm{Var}(X_k) = \frac{\alpha_k(\alpha_0 - \alpha_k)}{\alpha_0^2(\alpha_0 + 1)},
> \]$
>
> where $\alpha_0 = \sum_{k} \alpha_k$. The first two L-moments (used as robust analogues of mean and variance) serve as empirical estimators of these quantities. Substituting the empirical L-moments into the above identities and solving for $\alpha_k$ yields the closed-form expression in Eq.~(6).
>
> We have added a brief clarification in Appendix~A to make this connection explicit.
>
> W3: The ‘Random-Selection’ strategy applies only to the Epistemic Wrapper and does not influence the baseline INN. INN weights are initialized and trained in the standard way, so their accuracy in Table~1 is entirely independent of any budgeting choice.
>
> Budgeting determines which subset of posterior weights the wrapper transforms. Strategies such as High-$\sigma$ or High-$(\mu,\sigma)$ deliberately select weights whose posteriors show large epistemic spread, and these benefit the most from being converted into interval-valued belief representations. This is why such strategies yield strong improvements.
>
> Random-Selection, by contrast, samples weights uniformly from the entire posterior. It often misses the informative or highly uncertain weights that drive the benefit of wrapping, which explains the weaker performance. The INN baseline remains unchanged because it never uses budgeting.
>
> After fine-tuning, deeper networks (e.g., MLPs with $n=8$) perform well because the fine-tuning stage lets all weights adapt around the interval-based initialization introduced by the wrapper.

---

> > ### Author Response · Authors · 2025-11-22
> > **Revisions and clarifications addressing reviewer Minor remarks**
> >
> > W4: Responses to minor remarks,
> >
> > W4 (a):  We have increased the size of Figure 1 in the updated version to improve readability.
> >
> > W4 (b):  We have now added a clear definition of SPDE (Stochastic Partial Differential Equation) at its first occurrence in the paper.
> >
> > W4 (c):  We have corrected the misplaced boldface in Table 3.
> >
> > W4 (d):  We have corrected the misplaced boldface in Table 6.

---

> ### Comment · Reviewer_oFPB · 2025-11-22
>
> Thanks to authors for their detailed answers. I'll keep my score, but please take into account that the uncertainty estimation is quite far from my main research areas.

---

> > ### Author Response · Authors · 2025-11-24
> > **Updated PDF with highlighted revisions**
> >
> > **We have uploaded a revised version of the paper. All newly added or updated text is highlighted in GREEN. Please let us know if any further clarification is needed.**

---

### Official Review · Reviewer_Tv74 · 2025-10-30

**Soundness:** 2
**Presentation:** 3
**Contribution:** 2
**Rating:** 4
**Confidence:** 2

**Summary:**

This paper proposes Epistemic Wrapping, a method for improving uncertainty quantification in Bayesian Neural Networks (BNNs) by transforming parameter posteriors into belief-function posteriors. The idea is to model epistemic uncertainty in the parameter space rather than only in the output (prediction) space. The approach combines concepts from belief function theory, random set representations, and Dirichlet fitting, leading to interval-based inference via an Interval Neural Network (INN). The authors claim improved robustness, calibration, and out-of-distribution (OoD) detection on MNIST, CIFAR-10/100, and Fashion-MNIST benchmarks.

**Strengths:**

1. The idea of wrapping Bayesian posteriors in belief functions to capture higher-order epistemic uncertainty in the parameter space is quite novel (at least I haven't seen much).

2. The methodology is general well connected and nicely fit with each other.

3. Across datasets and architectures, this methods yields consistent accuracy and OOD gains.

**Weaknesses:**

1. While the method combines existing ingredients in an interesting way, several components (belief functions, Dirichlet evidential layers, random-set neural networks, interval networks) have all appeared in prior works. The authors also point out the novel lies in combing these techniques but I think then this is considered as rather incremental work maybe more suitable for a journal paper given nowadays AI conference requires more novel concepts.

2. With so much components is it hard to know which contributes the most to the final outcome, I know it might not make sense to ask the authors to do ablation on the components. But it is not clear which component contribute to what.

3. The writing of this paper is quite dense to parse, with many components referenced from existing work without explaining much of the details or intuition, which make it less accessible for the general audience.

**Questions:**

Please see my comments for the weakness section.

---

> ### Author Response · Authors · 2025-11-22
> **Revisions and clarifications addressing reviewer feedback**
>
> W1: We respectfully clarify that no prior work combines belief functions, Dirichlet modelling, and interval networks in the parameter space. Existing belief-function or evidential approaches model uncertainty in the target space, after predicting class probabilities [1, 2, 3, 4, 5]. Evidential Dirichlet layers similarly operate on logits or class outputs, not on weight posteriors.
> In contrast, our approach:
>
> (a) Transforms the BNN weight posterior into a continuous belief function over Borel intervals, using the likelihood-based construction of Wasserman (1990).
>
> (b) Represents these belief functions as random sets in parameter space, which to our knowledge has not been explored before in ML.
>
> (c) Fits a Dirichlet distribution to the resulting mass vectors, creating a second-order distribution over the Random-Set (RS) representation of parameters.
>
> (d) Performs inference using INNs, where the intervals naturally arise from the Dirichlet-wrapped parameters.
>
> This framework is therefore not a combination of existing modules, but a new methodological pipeline grounded in random-set theory and Bayesian consistency, enabling modelling of epistemic uncertainty at a level (parameter space) unexplored by prior approaches.
>
> We hope this clarification addresses the reviewer’s concern about incremental novelty
>
> References:
>
> [1] Sensoy, M., et al (2018),. Evidential Deep Learning.
>
> [2] Manchingal, S. et al. (2025). Random-set convolutional neural networks.
>
> [3] Wang, K. et al. (2025). Credal Interval Neural Networks.
>
> [4] Cuzzolin, F. (2020). The Geometry of Uncertainty.
>
> [5] Wasserman, L. (1990). Belief functions and statistical inference.
>
> W2: We agree that the Epistemic Wrapper contains multiple interconnected components, and we appreciate the opportunity to clarify their roles. Our pipeline is not a collection of independent modules but a mathematically linked sequence in which each step produces the representation required by the next: the BNN posterior provides a density; truncation defines a bounded support; belief and plausibility must be computed over closed intervals according to continuous random-set theory; Möbius inversion is the unique operator that converts these into basic probability masses; the resulting mass vector lies on a simplex and therefore requires a simplex-valued distribution such as the Dirichlet. The interval bounds derived from Dirichlet samples are necessary for propagation through an Interval Neural Network during inference, giving a coherent random-set representation in the parameter space. Because of this mathematical dependency, dropping individual components would break the representation entirely, making a standard “remove-one-block” ablation ill-defined.
>
> However, we would like to highlight that we have already included three ablation studies (Appendix Sec. C.1) targeting the modelling choices for which ablations are most meaningful:
>
> (Ablation 1) Distributional choices over the simplex:
> We compare three distributions for modelling the belief mass vectors. As shown in Appendix C.1 Figure 7, all three yield valid fits, but the Dirichlet offers the best balance of stability, interpretability, and consistency with epistemic-uncertainty literature.
>
> (Ablation 2) Number of closed intervals in belief-function construction:
> We vary the number of closed intervals used to compute belief values (10-60 intervals). As shown in Table 5 and Figure 8, the L-moment Dirichlet parameters stabilize around 30 intervals, after which changes become marginal. We therefore fix 30 intervals to achieve both stability and efficiency.
>
> (Ablation 3) Budgeting set size:
> We vary the fraction of wrapped weights from 5% to 50%. Even wrapping just 10% of the most informative weights yields a large gain over the baseline (9.33% to 45.34% before fine-tuning). After fine-tuning, performance is stable across budgeting levels (about 91-92%), with a slight peak at 30%. This shows that only a small, informative subset of weights is sufficient to obtain strong results.
>
> These findings confirm that the wrapper’s design choices are robust.
>
> W3: The Epistemic Wrapper integrates ideas from belief functions, random-set theory, L-moment Dirichlet modelling, and Interval Neural Networks. Each of these components comes from a substantial mathematical background, and providing full intuition for all of them in the main text is not feasible within the page limit. For this reason, we focused the main paper on the core pipeline and placed the extended explanations, definitions, and derivations in Appendix A.
>
> To improve accessibility, the revised version now clarifies the intuition behind each step directly in Section 3, and adds short guiding sentences before the mathematical constructions. We also included a high-level schematic summarizing the full pipeline. We believe these additions make the presentation lighter and easier to follow while still keeping the technical details available in the appendix.

---

> > ### Author Response · Authors · 2025-11-24
> > **Updated PDF with highlighted revisions**
> >
> > **We have uploaded a revised version of the paper. All newly added or updated text is highlighted in BLUE. Please let us know if any further clarification is needed**

---

### Meta-Review · Area_Chair_so7u · 2025-12-18

**Summary:**

The reviews for this paper are mixed. This paper was reviewed by four experts in the field and received 2 Reject (2), 1 Marginal Reject (4) and 1 Marginal Accept (6).

This article proposes “Epistemic Wrapping,” a new technique that aimed at improving uncertainty quantification in Bayesian neural networks by transforming the posterior distributions into belief function. This approach aims to model epistemic uncertainty directly in the parameter space rather than solely in the prediction space. It combines belief function theory, random set representations, Dirichlet fitting, and interval neural networks to produce interval-based inference from a trained BNN. The Epistemic Wrapping framework is applicable to any trained BNN and is evaluated on image classification benchmarks.
The reviewers largely agree that the article addresses an important and relevant issue and that the idea of “epistemic wrapping,” which combines belief functions, neural networks with intervals, and Bayesian a posteriori, is interesting and potentially promising. The article is generally well written.

Yet, based on the reviews, I side with the reviewers recommending rejection. The authors are encouraged to carefully consider the reviewers’ comments, to improve the paper for submission elsewhere.

**Reviewer Concerns:**

The reviewers raised several important concerns that can be summarized as follows:
1. Unclear novelty and positioning: Multiple reviewers questioned whether the contribution goes beyond an incremental combination of existing components (belief functions, Dirichlet evidential models, random sets, and interval neural networks). The novelty is primarily attributed to the integration of prior ideas rather than the introduction of a fundamentally new concept, which several reviewers felt is insufficient for a top-tier conference like ICLR.
2. Conceptual and motivational gaps: Reviewers expressed significant confusion regarding the core motivation, particularly the claim that existing methods model epistemic uncertainty only in the target space rather than the parameter space. This claim was considered misleading or incorrect, given that Bayesian neural networks explicitly model parameter uncertainty.
3. Soundness and theoretical clarity: Key methodological steps, such as the use of Dirichlet distributions to yield interval representations and the L-moment fitting procedure, seem to lack formal derivation or sufficient theoretical explanation. Also, it is hard to know which contributes the most to the final outcome.
4. Experimental reliability and competitiveness: Several reviewers raised serious concerns about the experimental setup. Reported baseline accuracies on standard benchmarks (e.g., MNIST, Fashion-MNIST, CIFAR-10) are below commonly reported state-of-the-art results, suggesting potential issues with training or evaluation.
5. Reproducibility and clarity of results: The lack of pseudocode or released code further makes reproducibility difficult.
6.  Interest of the technique: the proposed method does not make sense, the paper proposes to take a trained BNN, then transform the posterior distribution of the weights by dynamic truncation, then transform it into a belief function along intervals, then fit a dirichlet distribution using the L-moment method, then sample from this dirichlet distribution to obtain interval distributions and then build an interval neural network. The paper does not argue why these are good ideas

**Reviewer Scores:**

I don't think the two reviewers who gave a score of 2 would have increased their score. I note that the authors attempt to respond to reviewer pUPa. However, I agree that the notion of uncertainty in first-order parameters compared to second-order parameters is unclear. I have read the new paragraph (lines 93 to 106) in the main article, but I think that lines 59 to 91 have small issues (claims too strong) and I recommend completely rewriting the introduction.  Furthermore, I do not see how the DPsp reviewer could increase its score. I think that the justification for the interest of W5 techniques is not well answered. The idea proposed by the authors seems very incremental, and it is important to justify why this technique is important and what it brings to the community. The response does not make it clear what readers should read this paper. Furthermore, the low accuracy of the BNN is a little strange. I agree with the reviewers also on that point. I think the reviewers oFPB and Tv74 would not have changed their score. I think the answer to the W3 of reviewer Tv74
is a bit not sufficient. And this point is linked to W5, of reviewer DPsp. It seems that many Reviewers have difficulty understanding the interest of the techniques. Furthermore, if I had been an AC at the beginning of the article, I would also have interacted with the authors and asked them why they chose an old BNN architecture as BNN to be improve. Why not use a BNN based on Langevin dynamics or Hamiltonian Monte Carlo as in [1]? This seems to be one of the best techniques for estimating the posterior. Also I wonder how to handle a multimodal posterior as in [2]?


[1] Izmailov, P., Vikram, S., Hoffman, M. D., & Wilson, A. G. G. (2021, July). What are Bayesian neural network posteriors really like?. In International conference on machine learning(pp. 4629-4640). PMLR.

[2] Laurent, Olivier, Emanuel Aldea, and Gianni Franchi. "A symmetry-aware exploration of bayesian neural network posteriors." arXiv preprint arXiv:2310.08287 (2023).

---

### Decision · Program_Chairs · 2026-01-26

Reject